# Escaping the Mode: Multi-Answer Reinforcement Learning in LMs

**Isha Puri** [1]  **Mehul Damani** [1]  **Idan Shenfeld** [1]  **Marzyeh Ghassemi** [1]  **Jacob Andreas** [1]  **Yoon Kim** [1]

## Abstract

Given a question, a language model (LM) implicitly encodes a distribution over possible answers. In practice, post-training procedures for LMs often collapse this distribution onto a single dominant mode. While this is generally not a problem for benchmark-style evaluations that assume one correct answer, many real-world tasks inherently involve multiple valid answers or irreducible uncertainty. Examples include medical diagnosis, ambiguous question answering, and settings with incomplete information. In these cases, we would like LMs to generate multiple plausible hypotheses, ideally with confidence estimates for each one, and without (computationally intensive) repeated sampling to generate non-modal answers. This paper describes a *multi-answer* reinforcement learning approach for training LMs to perform distributional reasoning over multiple answers during inference. We modify the RL objective to enable models to explicitly generate multiple candidate answers in a single forward pass, internalizing aspects of inference-time search into the model's generative process. Across question-answering, medical diagnostic, and coding benchmarks, we observe improved diversity, coverage, and set-level calibration scores compared to single answer-trained baselines. Models trained with our approach require fewer tokens to generate multiple answers than competing approaches. On coding tasks, they are also substantially more accurate. These results position multi-answer RL as a principled and compute-efficient alternative to (externalized) inference-time scaling procedures like best-of-$k$. Code and more information can be found at `multi-answer-rl.github.io`.

[1]MIT CSAIL. Correspondence to: Isha Puri <ishapuri@mit.edu>.

*Proceedings of the 43rd International Conference on Machine Learning*, Seoul, South Korea. PMLR 306, 2026. Copyright 2026 by the author(s).

## 1. Introduction

Modern large language models (LMs) are typically (post-)trained via reinforcement learning to reason in natural language in order to produce a single best answer per query, implicitly incentivizing the most likely correct answer (Guo et al., 2025). This objective is fundamentally mismatched to many real-world settings in which multiple distinct answers may be simultaneously correct, or where uncertainty arises due to incomplete or ambiguous information. As a motivating example, consider a clinical setting in which a patient comes in complaining of *right lower quadrant abdominal pain and fever*. A clinician in this setting might suspect a diagnosis of *acute appendicitis* or a *right-sided kidney stone*, but be unsure as to which due to incomplete information (i.e., epistemic uncertainty over a single true diagnosis). To check, they might order a *complete blood count* and a *urinalysis*, both of which would be clinically appropriate tests to order (i.e., multiple correct answers).

For such applications, models should ideally be able to go beyond single answers and give a *set* of answers. However, in challenging tasks like math, coding, or medical diagnosis, LMs must typically generate a long natural-language *reasoning chain* before producing even a single correct answer. Training LMs to reason their way toward a single correct answer (e.g. using reinforcement learning) can often suppress alternative plausible hypotheses, leading models to repeatedly generate the same dominant answer even when other correct possibilities exist. Indeed, entropy collapse is a commonly documented failure mode of models trained via RL with binary rewards (Lin et al., 2025; Yu et al., 2025; Jin et al., 2025; Wu & Choi, 2025). Beyond requiring a set of answers, many applications of interest—especially in high-stakes settings—would further benefit from *uncertainty estimates* associated with each answer (Kapoor et al., 2024). In other words, we ideally want the response to be a *explicit distribution* over plausible responses, including their estimated probabilities.

Inference-time techniques have been proposed to address some of these limitations: techniques for sampling multiple answers through sampling in parallel (Puri et al., 2025; Beirami et al., 2025) or sequentially (Shinn et al., 2023; Xie et al., 2023) can produce an answer set for a given query; for uncertainty, models could be prompted or trained to verbal-

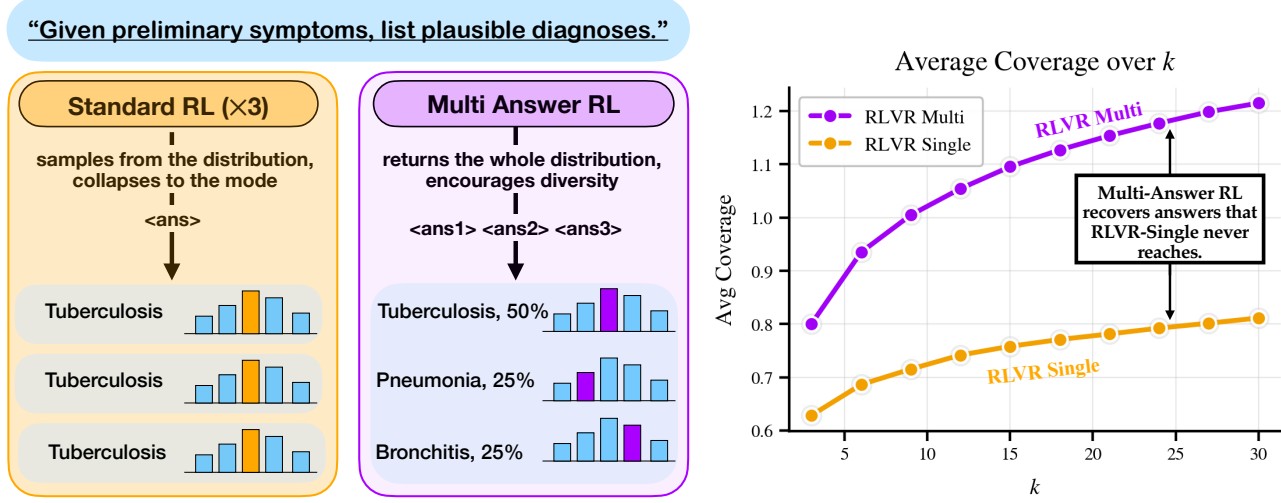

*Figure 1.* While standard RL trains LMs to consistently output the most likely answer to a question, **Multi-Answer Reinforcement Learning** trains models to output *distributions* of diverse answers.

ize uncertainty estimates for each answer (Xiong et al., 2024; Lin et al., 2022; Yang et al., 2025; Damani et al., 2025), or a separate model could be trained on top of model representations to output uncertainty scores (Liu et al., 2023; Azaria & Mitchell, 2023). However, these post-hoc methods do not change the underlying training objective, and this train-test mismatch could lead to poor performance in downstream decision-making scenarios where exploring and producing calibrated uncertainty estimates across multiple possibilities is crucial. This problem is particularly acute for models trained with modern post-training methods (as opposed to, say, base models), as while RLVR has made real strides in teaching models to reason, that training optimizes for a single high-reward completion, reasoning trace included. Recovering alternatives through repeated sampling then becomes both computationally expensive and behaviorally misaligned: the model has been trained to commit to one answer, not to hedge across several.

How can we move beyond single-answer RL objectives and instead train models to represent and generate sets of plausible answers directly? If diversity of responses and calibrated uncertainty estimates are desirable properties of language model outputs, then they should be explicitly optimized for during training rather than heuristically recovered at inference time. To this end, we propose **Multi-Answer RL**, an RL approach which explicitly optimizes language models to generate the distribution $P$ over answers directly.

Concretely, multi-answer RL trains models to reason jointly over multiple plausible hypotheses within a single chain of thought and to verbalize structured sets of candidate answers in a single generation. Our approach supports both single-answer and multi-answer ground-truth settings and encourages explicit hypothesis exploration rather than repeated

resampling. By introducing a reward function inspired by proper scoring rules that incentivizes calibrated answer distributions, we further show how this set generation approach can be extended to produce verbalized confidence scores for each answer in the set, thus making it into the full distribution $P$.

We empirically validate our approach on ambiguous and multi-label reasoning tasks (a general question answering dataset with incomplete information, a medical diagnostic dataset, and a coding dataset). We find that multi-answer RL scales better than baselines, producing substantial improvements in coverage, answer diversity, and token efficiency. For example, on coding tasks, our approach boosts top-1 accuracy by over 50% while cutting token usage by more than half. We further show that our set-level calibration reward enables models to produce better-calibrated uncertainty scores at the set level.

## 2. Background

We start by describing the standard reinforcement learning setting for language models. A policy parameterized as an LM $\pi_\theta$ maps a prompt $x \in X$ to a distribution over textual outputs $y \in Y$. Given a dataset $D = \{(x_i, y_i^*)\}$ of prompt–answer pairs and a reward function $R : Y \times Y \to \mathbb{R}$, training aims to maximize expected reward:

$$\max_\theta \; \mathbb{E}_{(x,y^*)\sim D, \; y\sim\pi_\theta(\cdot|x)} \big[ R(y, y^*) \big]. \tag{1}$$

This approach trains the LM to output samples which have high reward on average. In practice, this can result in policies that place much of its mass on single outputs.

**Reinforcement learning with verifiable rewards (RLVR).** RLVR focuses on the class of reward functions where rewards are deterministically verifiable from model outputs. A standard choice is the binary correctness reward,

$$R_{correct}(y, y^*) = \mathbb{1}_{y \equiv y^*}, \qquad (2)$$

where $\mathbb{1}_{y \equiv y^*} \in \{0, 1\}$ indicates whether the output $y$ matches the ground-truth answer $y^*$ (e.g. up to formatting differences).

**Reinforcement learning with calibration rewards (RLCR).** A key limitation of RLVR is that models are incentivized to guess when uncertain. Building on the theory of proper scoring rules (Gneiting & Raftery, 2007), RLCR prompts models to produce reasoning chains that output both an answer $y$ and an associated confidence estimate $q \in [0, 1]$ (Damani et al., 2025). Models are then trained using a reward that jointly incentivizes correctness and calibration:

$$R_{\text{RLCR}}(y, q, y^*) = \mathbb{1}_{y \equiv y^*} - S(q, \mathbb{1}_{y \equiv y^*}), \qquad (3)$$

where $S$ is a *proper scoring rule*. Damani et al. prove that for a particular class of proper scoring rules, RLCR incentivizes predictions are both accurate (in the sense that the highest-probability answer is generated) and calibrated. In practice, Damani et al. use the Brier score, which yields $S(q, \mathbb{1}_{y \equiv y^*}) = (q - \mathbb{1}_{y \equiv y^*})^2$.

## 3. Multi-Answer Reinforcement Learning

### 3.1. Multi-Answer RLVR

We consider a generalized setting in which a prompt $x$ is associated not with a single ground-truth answer, but with a *set* of valid answers $\mathcal{Y}^*(x) = \{y_1^*, y_2^*, \ldots, y_N^*\}$, where $N \geq 1$ may vary across instances. Our goal is to train a model to recover the full set—or a high-coverage subset—of these valid answers within a single generation.

We prompt the policy $\pi_\theta$ to produce a structured output consisting of a set of $K$ *distinct* candidate answers,

$$A = \{a_1, a_2, \ldots, a_K\},$$

within a single chain of thought. We then train the model using a *set-level* reward that checks how many of the generated answers belong to the ground-truth set:

$$R_{\text{RLVR}}^{\text{multi}}(A, \mathcal{Y}^*) = \sum_{i=1}^{K} \mathbf{1}[a_i \in \mathcal{Y}^*], \qquad (4)$$

This objective can be viewed as a natural generalization of RLVR from single-answer correctness to set-level correctness and subsumes several familiar training objectives:

1. $N = 1$, $K = 1$ (**Standard RLVR**). When there is a single ground-truth answer and the model produces a single output, the reward reduces exactly to the binary correctness signal used in vanilla RLVR.

2. $N = 1$, $K > 1$ (**Pass@$K$**). When there is a single correct answer but the model produces multiple candidates, the reward is equivalent to a pass@$K$ objective.

3. $N > 1$, $K \leq N$ (**Partial set recovery**). When multiple correct answers exist but the model is constrained to produce fewer candidates than the size of the ground-truth set, the objective encourages maximal coverage of distinct valid answers.

4. $N > 1$, $K \geq N$ (**Full set recovery**). When allowed to generate at least as many candidates as valid answers, the optimal policy recovers the entire ground-truth set.

Since we want to output distinct answers from the model, we use an additional *format reward* to enforce uniqueness among the generated answers. Full details are in Appendix C.

### 3.2. Multi-Answer RLCR

Building on Multi-Answer RLVR, we introduce *Multi-Answer RLCR*, which additionally trains models to produce calibrated confidence estimates per answer. In a single chain-of-thought, the model is asked to output both a set of $K$ distinct candidate answers $A = \{a_1, a_2, \ldots, a_K\}$ and a corresponding set of confidence values $Q = \{q_1, q_2, \ldots, q_K\}$ where each $q_i \in [0, 1]$ represents the model's reported confidence that the answer $a_i$ is correct.

As in RLCR, training jointly optimizes answer correctness and confidence calibration by combining a correctness-based reward with a proper scoring rule. To measure calibration in the multi-answer setting, we use the *Multi-Brier score*, defined as the average squared error between reported confidences and answer-level correctness:

$$R_{\text{Brier}}^{\text{multi}}(A, Q, \mathcal{Y}^*) = \frac{1}{K} \sum_{i=1}^{K} \Big( q_i - \mathbf{1}[a_i \in \mathcal{Y}^*] \Big)^2.$$

Intuitively, this objective trains each confidence $q_i$ to approximate the probability that the specific answer $a_i$ is correct. The full objective, which jointly incentivizes coverage of valid answers and calibrated confidence estimates at the answer level, is given by

$$R_{\text{RLCR}}^{\text{multi}}(A, Q, \mathcal{Y}^*) = R_{\text{RLVR}}^{\text{multi}}(A, \mathcal{Y}^*) - R_{\text{Brier}}^{\text{multi}}(A, Q, \mathcal{Y}^*).$$

This recovers RLCR (Damani et al., 2025) as a special case when $N = K = 1$.

> **Condensed RLCR-Multi Example Output**
>
> ```
> <think>Tuberculosis is a consideration, [...] then
> there's pneumonia, which can cause similar
> symptoms. [...] Another possibility is bronchitis,
> but again, hemoptysis is more common in other
> conditions. [...] Given the hemoptysis and fever,
> TB and pneumonia are higher. Since the patient is
> underweight, TB might be more likely as
> undernutrition can impair immune response. [...]
> Bronchitis is possible but less so. [...] ranking TB
> first, pneumonia second, bronchitis third[...]
> Assigning confidences: TB maybe 0.4, pneumonia
> 0.30, bronchitis 0.30.</think>
> ```
>
> ```
> <a1>Tuberculosis</a1><conf1>0.40</conf1>
> ```
>
> ```
> <a2>Pneumonia</a2><conf2>0.30</conf2>
> ```
>
> ```
> <a3>Bronchitis</a3><conf3>0.30</conf3>
> ```

**Model output as a distribution.** The model's output $(A, Q)$ can be interpreted as defining a distribution over plausible answers. When $N = 1$, our format reward enforces that the $K$ confidence scores sum to at most one, yielding a discrete probability distribution over the answer set. When $N > 1$, confidence scores do not necessarily need to sum to one, and the output naturally corresponds to a multivariate Bernoulli distribution over correctness events.

This approach is also closely related to **conformal prediction** approaches (Angelopoulos & Bates, 2023). In our approach however models learn to output the $K$ most likely answers from $p(y \mid x)$ rather than guaranteeing that these answers will cover some fixed fraction of its probability mass.

## 4. Experiments

### 4.1. Experimental Setup

**Training Details.** We use GRPO as the base RL algorithm with some modifications (see Appendix C). Specifically, we use the Qwen3-8B base models with a max response length of 1536. To enable easier verification and more structured outputs, we augment our reward with a simple format reward that encourages models to enclose CoTs within the right tags and keep answers produced in Multi-Answer sets unique.

**Datasets.** We evaluate our approach on three datasets (DDXPLUS, HOTPOTQA-MODIFIED, MBPP) that require set-valued reasoning, but differ in structure: DDXPlus admits multiple simultaneously correct answers given incomplete medical information; MBPP consists of well-specified, unambiguous tasks that admit multiple correct implementations via distinct algorithmic approaches; and HotPotQA-Modified has a single gold answer but significant ambiguity

due to incomplete or underspecified context.

**1. DDXPlus** (Tchango et al., 2022) is a large-scale medical diagnostic dataset in which each example consists of basic patient demographics along with a brief description of symptoms and antecedents. The target output is a *differential diagnosis*—a *set* of medical conditions that are plausible given the available information. Because multiple diagnoses may be simultaneously correct ($N \geq 1$), performance is naturally evaluated using coverage over the set of gold answers rather than top-1 accuracy. A representative example is shown in Figure 1. We train on 25,000 examples, evaluate correctness using exact string match, and prompt models to generate $K = 3$ diagnoses in a single output.

**2. HotPotQA-Modified** is a modified version of the HotPotQA distractor dataset (Yang et al., 2018). Each example contains a multi-hop question with 10 paragraphs (2 relevant, 8 distractors), where we remove 1 or both relevant paragraphs to vary information completeness. Although each question retains a single ground-truth answer ($N = 1$), incomplete information introduces ambiguity, making it beneficial for models to reason over and output multiple plausible candidates. This setting corresponds to the $N = 1$, $K > 1$ regime, where the Multi-Answer RL objective corresponds to a pass@$K$ reward.

**3. Coding: MBPP** (Austin et al., 2021) is a benchmark of crowd-sourced programming tasks, each with a natural language description and unit tests, and is well-specified and unambiguous. While multiple implementations may be correct, they correspond to distinct algorithmic approaches for solving the same underlying task, rather than different valid answers. We measure the number of unique answers using AST-based uniqueness (i.e., answers are considered distinct if their abstract syntax trees differ). We use MBPP to represent the *low-ambiguity, multi-solution* regime, enabling evaluation of Multi-Answer RL across the full spectrum—from inherently multi-answer (DDXPlus), to ambiguous single-answer (HotPotQA-Modified), to structured code generation tasks.

**Methods.** We evaluate the following methods:

1. **Base**: The base pre-trained model (Qwen3-8B). In our experiments, we prompt the base model with both single and multi versions of RLVR and RLCR prompts.

2. **RLVR Single:** Initialized from the base model and trained using standard RLVR to output a single answer. During evaluation, the model is also prompted to output a verbalized confidence.

3. **RLCR Single:** Initialized from the base model and trained using RLCR (Preliminaries) to output a single answer and corresponding confidence.

| Method | Set ECE | T1 ECE | Tk ECE | B1 Brier | Bk Brier |
|---|---|---|---|---|---|
| **DDXPlus: Medical** | | | | | |
| RLVR-S | – | 0.34 | – | 0.35 | – |
| RLCR-S | – | 0.02 | – | **0.24** | – |
| RLVR-M | 0.13 | 0.16 | 0.10 | 0.27 | 0.19 |
| RLCR-M | **0.02** | **0.01** | **0.04** | **0.24** | **0.18** |
| **HotPotQA-Hard: Trivia** | | | | | |
| RLVR-S | – | 0.48 | – | 0.38 | – |
| RLCR-S | – | **0.13** | – | **0.16** | – |
| RLVR-M | 0.47 | 0.38 | 0.22 | 0.30 | 0.16 |
| RLCR-M | **0.44** | 0.31 | **0.21** | 0.22 | **0.12** |
| **MBPP: Coding** | | | | | |
| RLVR-S | – | 0.56 | – | 0.55 | – |
| RLCR-S | – | 0.53 | – | 0.53 | – |
| RLVR-M | 0.44 | 0.54 | 0.51 | 0.51 | 0.48 |
| RLCR-M | **0.26** | **0.37** | **0.33** | **0.38** | **0.34** |

*Table 1.* **Calibration metrics for Multi-Answer RL ($k = 3$).** Lower is better. S and M denote Single and Multi. T1/Tk denote Top-1/Top-$k$.

4. **RLVR Single Prompted with RLVR Multi Prompt:** We prompt the trained RLVR Single model to produce multiple answers using RLVR Multi prompt.

5. **RLCR Single Prompted with RLCR Multi Prompt:** We prompt the trained RLCR Single model to produce multiple answers and confidences using the RLCR Multi system prompt.

6. **RLVR Multi (ours):** Trained with $R_{\text{RLVR\_Multi}}$. Answers are extracted from `<answer{i}>` tags.

7. **RLCR Multi (ours):** Trained with $R_{\text{RLCR\_Multi}}$. Answers and confidences are extracted from `<answer{i}>` and `<confidence{i}>` tags.

We compare Single-Answer and Multi-Answer methods by constructing sets of $K$ candidate answers in both cases. For Single models, we sample $K$ independent responses and treat them as a set, while Multi models naturally produce a single set of $K$ answers in a single generation. This comparison allows us to isolate the effect of multi-answer training from inference-time sampling.

**Correctness Metrics.** We evaluate correctness, diversity, and efficiency using the following metrics:

1. **Coverage (Avg. # Correct per Set) (↑):** Measures the average number of correct answers produced per example. Formally, for a set of $K$ answers,

$$\text{Coverage } = \frac{1}{K}\sum_{i=1}^{K}\mathbb{1}\{a_i \text{ is correct}\}. \quad (5)$$

2. **Pass@1 (↑):** Measures the accuracy of a single selected answer from the generated set. For multi-answer

methods, we use the first answer in the set, which we empirically find the model treats as most likely (implicitly in the RLVR case, explicitly in the RLCR case). For single-answer RLVR with independent samples, we select an answer uniformly at random from the set.

3. **Avg. Token Count (↓):** The average, over all questions, of the total token count across $K$ generated answers.

4. **Uniqueness (↑):** The number of distinct answers within the generated set, measuring output diversity.

**Calibration Metrics.** All methods are prompted to output a confidence score $q_i \in [0, 1]$ for each generated answer $a_i$. We evaluate calibration using the following metrics:

1. **Brier Score (↓).** Measures the squared error between predicted confidence and binary correctness:

$$\text{Brier} = \frac{1}{K}\sum_{i=1}^{K}\big(q_i - \mathbb{1}\{a_i \in \mathcal{Y}^*\}\big)^2. \quad (6)$$

We report Brier scores for the top-ranked answer (used for Pass@1) and for all answers pooled across the set.

2. **Expected Calibration Error (ECE) (↓).** Measures the discrepancy between predicted confidence and empirical accuracy by binning confidence scores:

$$\text{ECE} = \sum_{m=1}^{M}\frac{|B_m|}{L}\left|\text{acc}(B_m) - \text{conf}(B_m)\right|, \quad (7)$$

where $M$ is the number of bins, $B_m$ denotes the samples in bin $m$, and $L$ is the total number of samples. We use $M = 10$, and report ECE both for the top-ranked answer and the entire set of answers.

3. **Set ECE (↓, diagnostic).** Measures calibration at the level of answer sets. For each example, we define set-level correctness $y_{\text{set}} = \mathbb{1}\{\exists i : a_i \in \mathcal{Y}^*\}$. For datasets where questions only admit one correct answer (and thus set level probabilities must sum to 1, we define set-level confidence $q_{\text{set}} = \sum_{i=1}^{K} q_i$. When there are multiple correct answers and probabilities can sum to $> 1$, set-level confidence is defined as $q_{\text{set}} = 1 - \prod_{i=1}^{K}(1 - q_i)$. Set ECE is computed by binning $q_{\text{set}}$ and comparing empirical set accuracy to predicted set confidence, much like standard ECE.

**Comparability Across Settings.** Answer sets produced by single-answer and multi-answer methods differ in a fundamental way. Single-answer methods construct sets via repeated sampling, which often results in duplicate answers and variable effective set sizes. In contrast, multi-answer methods explicitly generate a set of distinct candidates. Because of this mismatch, **pooled** and **set-level** calibration

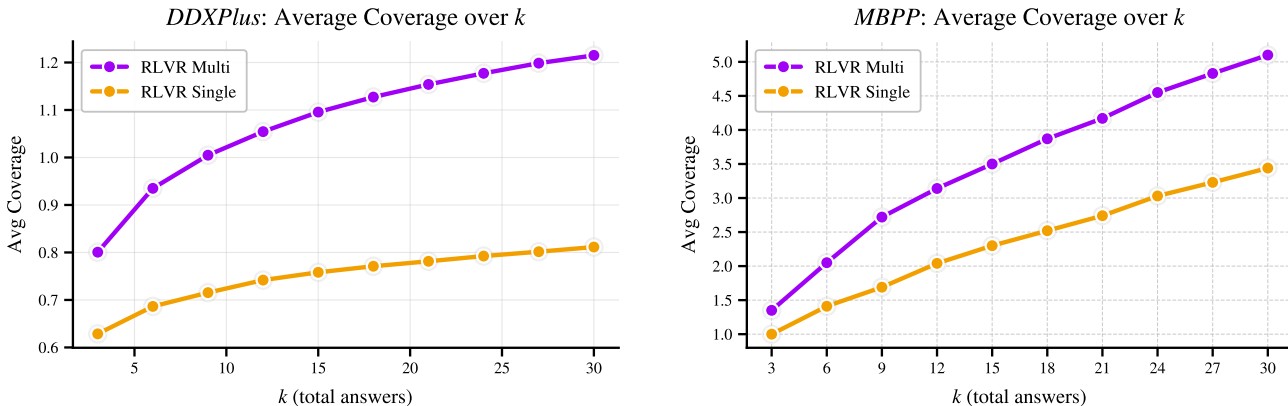

*Figure 2.* On DDXPLUS (left) and MBPP (right), we generate 30 answers from RLVR - 30 individual samples from RLVR Single, and 10 sets of 3 from RLVR Multi. Despite equal total generations, RLVR-Multi produces significantly more unique correct answers than RLVR-Single, indicating that RLVR-Single's mode-seeking behavior limits diversity.

metrics are not directly comparable across single and multi settings, as their values depend on set size. Accordingly, we use pooled and set-level calibration metrics only to compare *RLVR Multi* and *RLCR Multi*, where the answer set construction mechanism is shared. In contrast, **Top-1 Individual ECE** and **Top-1 Brier Score** evaluate calibration of the highest-ranked answer, and are directly comparable.

### 4.2. Main Results

In this section, we compare *Single-Answer* to *Multi-Answer* RL, as well as *RLVR* to *RLCR*, along both correctness and

calibration on DDXPLUS, HOTPOTQA-MODIFIED, and MBPP.

**Correctness.** Table 2 reports correctness results across all settings. Across both RLVR and RLCR, Multi-Answer models substantially outperform their Single-Answer counterparts on set-level correctness metrics including coverage. Moreover, simply prompting single-answer models to produce multiple answers performs substantially worse than trained multi-answer models, showing that generating answer sets requires explicit multi-answer training.

| (a) DDXPlus: Medical Differential Diagnoses | | | | |
|---|---|---|---|---|
| **Method** | **Avg. Corr. ↑** | **Div. ↑** | **Eff. ↓** | **Top-1 ↑** |
| RLVR-Single | 0.62 | 0.62 | 1191 | **0.50** |
| RLVR-Single + Multi | 0.67 | 0.99 | 695 | 0.42 |
| Zero-Shot Multi | 0.31 | 0.93 | 1131 | 0.24 |
| **Multi-Answer RLVR** | **0.79** | **1.00** | **677** | 0.42 |
| RLCR-Single | 0.65 | 0.50 | 1378 | **0.55** |
| RLCR-Single + Multi | 0.49 | 1.00 | 703 | 0.32 |
| Zero-Shot Multi | 0.41 | 0.89 | 1269 | 0.16 |
| **Multi-Answer RLCR** | **0.77** | **1.00** | **510** | 0.43 |
| (b) HotPotQA-Modified | | | | |
| **Method** | **Avg. Corr. ↑** | **Div. ↑** | **Eff. ↓** | **Top-1 ↑** |
| RLVR-Single | 0.17 | 0.59 | 1466 | **0.19** |
| RLVR-Single + Multi | 0.23 | 1.00 | 551 | 0.17 |
| Zero-Shot Multi | 0.20 | 0.75 | 1265 | 0.17 |
| **Multi-Answer RLVR** | **0.27** | **1.00** | **544** | **0.19** |
| RLCR-Single | 0.22 | 0.62 | 1782 | 0.17 |
| RLCR-Single + Multi | 0.23 | 0.98 | 670 | 0.17 |
| Zero-Shot Multi | 0.20 | 0.73 | 1298 | 0.15 |
| **Multi-Answer RLCR** | **0.27** | **1.00** | **622** | **0.19** |
| (c) Coding: MBPP | | | | |
| **Method** | **Avg. Corr. ↑** | **Div. ↑** | **Eff. ↓** | **Top-1 ↑** |
| RLVR-Single | 0.98 | 2.09 | 511.73 | 0.29 |
| RLVR-Single + Multi | 0.95 | 1.93 | 695.13 | 0.35 |
| Zero-Shot Multi [RLVR] | 0.91 | 1.94 | 664.78 | 0.34 |
| **Multi-Answer RLVR** | **1.35** | **2.98** | **235.49** | **0.49** |
| RLCR-Single | 0.89 | 2.13 | 518.07 | 0.27 |
| RLCR-Single + Multi | 0.92 | 1.90 | 724.89 | 0.35 |
| Zero-Shot Multi [RLCR] | 0.97 | 2.02 | 677.93 | 0.37 |
| **Multi-Answer RLCR** | **1.38** | **2.94** | **250.94** | **0.48** |

*Table 2.* **Correctness, diversity, and efficiency for Multi-Answer RL.** Multi-Answer RL substantially improves coverage, diversity, and efficiency over single-answer baselines on DDXPlus **(a)**, HotPotQA-Modified **(b)**, and MBPP **(c)**. All results use $k = 3$.

On DDXPLUS, where multiple diagnoses may be simultaneously correct, coverage is the primary notion of correctness. Multi-Answer models recover a significantly larger fraction of the gold differential diagnoses per example, demonstrating that explicitly optimizing for set-valued outputs enables models to capture relevant alternatives that might be suppressed by single-answer RL. A similar trend is observed on MBPP, where Multi-Answer models produce a more diverse set of correct implementations, capturing distinct algorithmic approaches that are often missed under single-answer training.

On HOTPOTQA-MODIFIED, each question admits a single correct answer but missing information induces significant uncertainty. In this setting, Multi-Answer RL yields large gains in pass@$k$, reflecting improved coverage of plausible answers. This improvement arises despite using just a single generation: by reasoning over multiple plausible hypotheses within a single generation, Multi-Answer models are more likely to include the correct answer among their candidates. In contrast, Single-Answer models exhibit answer collapse, repeatedly regenerating the same dominant answer, limiting coverage and under performing on pass@$k$.

**Calibration.** Table 1 reports calibration results across RLVR/RLCR Single/Multi. Across all three datasets, RLCR consistently improves calibration relative to RLVR in both Single- and Multi-Answer settings.

On DDXPLUS and MBPP, RLCR-Multi achieves markedly better set-level calibration than RLVR-Multi across all metrics, demonstrating RLCR's ability to assign meaningful confidences to multiple plausible answers. Calibration curves (Figure 3) provide a complementary view for DDX-PLUS. RLCR-Multi is better calibrated than RLVR-Multi overall, though it still deviates from perfect calibration, especially at higher confidence levels. In contrast, RLVR-Multi exhibits systematic overconfidence across the range.

RLCR-Multi closely tracks the identity line across confidence bins, while RLVR-Multi exhibits systematic overconfidence, particularly at higher confidence levels—effects that are less visible in aggregate metrics such as ECE.

Finally, of the calibration metrics considered, only **Top-1 ECE** and **Top-1 Brier** permit direct comparison between Single- and Multi-Answer RL. These metrics show that Multi-Answer training matches Single-Answer RL in calibration, indicating that explicitly optimizing for set-valued outputs does not degrade calibration of the top answer.

On HOTPOTQA-MODIFIED, the quantitative trends observed on DDXPLUS largely persist, with RLCR improving calibration relative to RLVR. Surprisingly, however, while

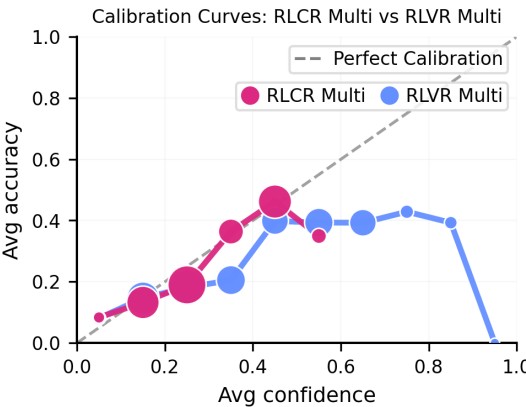

*Figure 3.* Calibration curves on DDXPlus. RLCR-Multi is significantly better calibrated than RLVR-Multi, though it diverges at higher confidences. RLVR-Multi remains systematically overconfident. The size of each dot corresponds to how many examples are found in that bucket.

it outperforms Multi-Answer RLVR, Multi-Answer RLCR underperforms Single-Answer RLCR, indicating degraded calibration. On inspection, we find that this behavior stems from the model having a strong generative prior to sum predicted confidences across the answer set to 1. In a challenging single–gold-answer setting such as HOTPOTQA-MODIFIED (pass@$K < 30\%$), well-calibrated behavior would instead assign total confidence strictly less than one for most questions. While, in principle, exploration during RL training could overcome this prior, we find that this does not occur in practice. We leave the development of improved exploration strategies for future work.

### 4.3. Analysis

**Does multi-answer RL improve output diversity?** Many tasks admit multiple plausible answers, making it important for models to explore a broader set of hypotheses rather than collapsing to a single dominant mode. While Table 2 showed that multi-answer training improves coverage, indicating that models generate a wider range of plausible answers, we analyze diversity more directly here. To analyze diversity more directly, we compare RLVR-Single and RLVR-Multi under a matched sampling budget on both the medical domain (DDXPLUS) and coding (MBPP). We sample 30 generations from RLVR-Single, yielding 30 answers, and 10 generations from RLVR-Multi, yielding $10 \times K = 30$ answers total. We then compute the number of unique answers per set and plot the distribution in Figure 4. We also plot average coverage—the mean number of correct answers per set—as a function of $K$ in Figure 2 for both DDXPLUS and MBPP.

Across RLVR-Multi exhibits greater output diversity, producing nearly twice as many unique answers on average (8 vs. 4). Moreover, its higher average coverage indicates

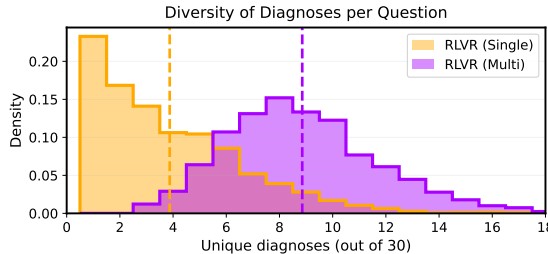

*Figure 4.* Distribution of the number of unique diagnoses per question across 5,000 test examples of DDXPLUS. RLVR-Multi produces more distinct diagnoses than RLVR-Single, explaining coverage gains under multi-answer training.

that this increased diversity corresponds to a larger number of correct answers, rather than spurious variation. Taken together, these results demonstrate that multi-answer training increases diversity without compromising correctness, addressing a key failure mode of single-answer RL.

**Is multi-answer RL more compute efficient than single-answer RL?** The standard approach to obtaining answer sets from single-answer models is inference-time scaling, most commonly by repeated sampling. Figure 5 illustrates a representative instance, showing three sampled generations from an RLVR-Single model. Although each generation produces a different final answer, the corresponding chains exhibit substantial overlap, repeatedly reproducing the same reasoning scaffolding and intermediate phrases. Quantitatively, this manifests as high $n$-gram overlap across samples (see Figure 5), indicating that inference-time sampling often incurs significant computational redundancy.

To compare this redundancy against multi-answer training, we measure the number of tokens for a set of $K$ answers under both approaches. For RLVR-Single, we sample the model $K$ times independently and sum the resulting token lengths. For RLVR-Multi, we generate $K$ answers within a single response and measure its total token length. In the medical domain, the average token length of an RLVR-Multi response is only $56\%$ of the total token length required by an RLVR-Single model to produce the same number of answers (Figure 7). These results show that training models to directly generate answer sets can eliminate a significant amount of redundant computation incurred by inference-time sampling.

**Summary.** Together, these results show that training models to explicitly optimize multi-answer structure results in outputs that are more accurate, calibrated, and diverse, while simultaneously being more efficient to generate.

## 4.4. Does multi-answer training remain stable with increasing K?

To understand the performance of multi-answer training as $K$ increases, we train Multi-Answer RLVR with $K = \{2, 3, 4, 5\}$. Because each medical example has multiple correct diagnoses, increasing $k$ directly expands the model's capacity to recover valid alternatives. As shown in Figure 6, the average number of correct answers per set increases monotonically with $k$. Since outputs are constrained to be unique, these gains reflect the model surfacing additional correct hypotheses rather than repeating dominant answers. Training remains stable across all values of $K$.

## 5. Related Work

**RLVR and Diversity.**

Recent work has documented systematic trade-offs between RL training and output diversity (Kirk et al., 2024; Shypula et al., 2025; Yang & Holtzman, 2025). (Wu & Choi, 2025) and (Yue et al., 2025) showed that improvements in pass@1 under RLVR are accompanied by a contraction of the model's support, narrowing its reasoning space and leading to degraded pass@k performance. Similarly, (West & Potts, 2025) find that base models often outperform aligned models on creative tasks. Several works have proposed methods to preserve diversity during RL training. (Chen et al., 2025) and (Walder & Karkhanis, 2025) directly optimize pass@k-style objectives by rewarding responses based on their contribution to set-level success. (Li et al., 2025) augment standard quality rewards with a learned diversity reward. (Song et al., 2025) use exploration bonuses conditioned on final outcomes. A complementary line of work focuses on explicitly controlling policy entropy during training (Cui et al., 2025; Cheng et al., 2025). Most of these approaches operate in the single-answer setting, focusing on encouraging diversity rather than training models to explicitly generate answer sets within a single generation.

**Calibration.** Reliable deployment of language models requires accurate uncertainty estimation (Kalai et al., 2025). Prior work has explored a range of approaches for estimating model confidence, including intent probing (Kadavath et al., 2022; Gupta et al., 2024; Azaria & Mitchell, 2023), sampling-based uncertainty estimation (Kuhn et al., 2023; Kang et al., 2025), and post-hoc verbalization (Xiong et al., 2024; Lin et al., 2022). Among these, post-hoc verbalization, where models output a confidence score after generating a response, has recently gained popularity. However, multiple works have shown that models are systematically overconfident when verbalizing their confidence (Xiong et al., 2024; Mei et al., 2025; Kirichenko et al., 2025). To address this, recent work has focused on directly training models to produce calibrated confidence estimates. (Lin et al., 2022)

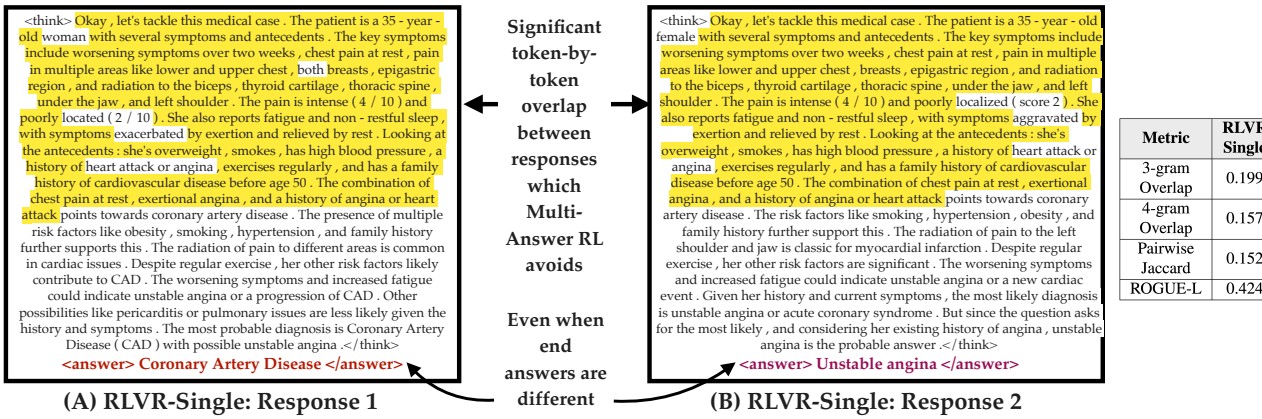

**(A) RLVR-Single: Response 1**    **(B) RLVR-Single: Response 2**

*Figure 5.* **Significant subsequence overlap between independently sampled RLVR-Single responses, even those that yield different answers,** indicating that independent sampling largely re-instantiates the same reasoning tokens. Multi-Answer RL mitigates this effect by optimizing multiple generations jointly, which reduces repeated token sequences and yields lower within-question overlap.

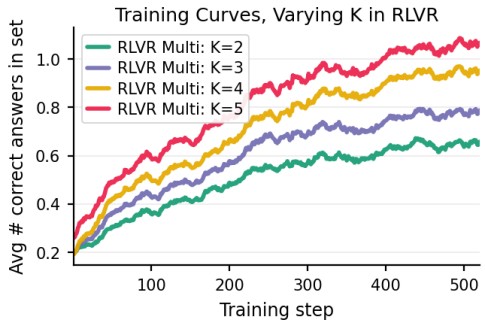

*Figure 6.* As $k$ increases, Multi-Answer RLVR stably recovers more unique correct diagnoses per set.

fine-tune GPT-3 to predict confidence conditioned on a question and generated answer. A smaller line of work uses RL with proper scoring rules (Brier, 1950; Gneiting & Raftery, 2007) as training rewards to align predicted confidence with empirical correctness. Within this paradigm, (Stangel et al., 2025) and (Xu et al., 2024) optimize purely for calibration, whereas (Damani et al., 2025) employ a joint objective that incentivizes both correctness and calibration.

**Generating Answer Sets.** There have been several recent efforts aimed at eliciting answer sets from LMs. (Wang et al., 2024) prompt models to output probability distributions over a fixed set of labels. (Zhang et al., 2025) introduce verbalized sampling, a prompting strategy that asks models to explicitly verbalize a distribution over multiple responses. Similarly, (Troshin et al., 2025) show that prompting models to enumerate or iteratively sample answers can increase output diversity. Although similar in intuition, these approaches are training-free and operate purely at inference time. Most closely related to our work is concurrent research by (Wang et al., 2025), which extends RLCR (Damani et al., 2025) to train models to output verbalized probability distributions over answer sets. Although aligned in motivation, their primary focus is on calibration of answer distributions,

whereas we study the broader benefits of multi-answer training, including diversity (pass@K) and computational efficiency.

## 6. Conclusion

We propose Multi-Answer Reinforcement Learning for language models, which trains reasoning LMs to generate sets of plausible answers rather than a single most probable response. Across medical and QA benchmarks, this objective improves coverage and diversity, recovering correct alternatives missed by standard RL while using fewer total tokens. More broadly, our work extends a line of approaches that use language to serialize structured reasoning, moving from single answers to explicit representations of output distributions. Limitations remain: single-answer objectives still achieve higher top-1 accuracy, our experiments are confined to QA, and serial generation limits parallelism despite improved token efficiency. Nevertheless, these results suggest a path toward models that can explore their full internal distribution without sacrificing performance or calibration.

## Impact Statement

This paper presents work whose goal is to advance the field of machine learning. There are many potential societal consequences of our work, none of which we feel must be specifically highlighted here.

## Acknowledgments

This research was supported by the MIT-IBM Watson AI Lab and the DARPA AIQ program through the DARPA CMO contract number HR00112520025. IP is supported by the NSF Graduate Research Fellowship. JA is additionally supported by a Sloan Research Fellowship. MG is supported in part by National Science Foundation (NSF) 22-586 Fac-

ulty Early Career Development Award (#2339381) and the AI2050 Program at Schmidt Sciences. YK was supported in part by the National Science Foundation under CAREER Award No. 2441872.

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

## A. Main Example

---
**RLVR-Single vs. RLVR-Multi: MBPP Output Diversity Example**

**Prompt:**    `Write a python function to check whether the given two numbers have same number of digits.`

**RLVR-Multi (3 samples):**

```
[Candidate 1]
def same_number_of_digits(num1, num2):
    len1 = len(str(num1))
    len2 = len(str(num2))
    return len1 == len2

[Candidate 2]
def same_number_of_digits(num1, num2):
    digits1 = 0
    digits2 = 0
    while num1 > 0:
        num1 //= 10
        digits1 += 1
    while num2 > 0:
        num2 //= 10
        digits2 += 1
    return digits1 == digits2

[Candidate 3]
def same_number_of_digits(num1, num2):
    str_num1 = str(num1)
    str_num2 = str(num2)
    return len(str_num1) == len(str_num2)
```

**RLVR-Single (3 samples):**

```
[Candidate 1]
def same_number_of_digits(num1, num2):
    return len(str(num1)) == len(str(num2))

[Candidate 2]
def same_number_of_digits(a, b):
    return len(str(a)) == len(str(b))

[Candidate 3]
def same_number_of_digits(num1, num2):
    return len(str(num1)) == len(str(num2))
```

*RLVR-Single repeatedly produces nearly identical programs, while RLVR-Multi generates functionally correct but structurally diverse implementations.*

---

## B. System Prompts

.

---
**RLCR Single Prompt**

```
A conversation between User and Assistant.  The user asks a question, and the
Assistant solves it.  The assistant first thinks about the reasoning process in the
mind, provides the user with the final answer, then analyzes its confidence about
the solution and then provides the user with its confidence level.  The confidence
level is a number between 0 and 1 (inclusive) enclosed within <confidence>
</confidence> tags.  The final answer is enclosed between <answer> </answer>
tags.  The analysis about confidence and uncertainty is enclosed within <analysis>
</analysis> tags.  Here are some guidelines for the analysis: 1.  Your task is to
point out things where the model could be wrong in its thinking, or things where
there might be ambiguity in the solution steps, or in the reasoning process itself.
2.  You should not suggest ways of fixing the response, your job is only to reason
about uncertainties.  3.  For some questions, the response might be correct.  In
these cases, it is also okay to have only a small number of uncertainties and then
explicitly say that I am unable to spot more uncertainties.  4.  Uncertainties might
be different from errors.  5.  If there are alternate potential approaches that
may lead to different answers, you should mention them.  6.  List out plausible
uncertainties, do not make generic statements, be as specific as possible.  7.
Enclose this uncertainty analysis within <analysis> </analysis> tags.  The final
format that must be followed is:  <think> reasoning process here </think> <answer>
final answer here </answer> <analysis> analysis about confidence and uncertainty
here </analysis> <confidence> confidence level here (number between 0 and 1)
</confidence>
```

---

.

---

**RLVR Single Prompt**

```
A conversation between User and Assistant.  The user asks a question, and the
Assistant solves it.  The assistant first thinks about the reasoning process in the
mind and then provides the user with the answer.  The reasoning process and answer
are enclosed within <think> </think> and <answer> </answer> tags, respectively, i.e.,
<think> reasoning process here </think><answer> answer here </answer>.  Do NOT put
any sentences or reasoning process within the <answer> </answer> tags – only put
the final answer that will be verified with exact match score within the <answer>
</answer> tags.
```

.

---

**RLCR Multi Prompt**

```
A conversation between User and Assistant.  The user asks a question, and the
Assistant solves it.  The question may be ambiguous or difficult to answer, and
you must propose multiple possible answers.  You must assign a confidence score
to each candidate.  Make sure the confidences sum to less than or equal to 1.  The
confidences are allowed to sum to less than 1 if you are unsure about all candidates.
You will be graded on how closely your confidences match the actual correctness
of the candidates.  Output EXACTLY {K} DISTINCT candidates with confidences that
sum to less than or equal to 1.  FORMAT ONLY (no extra text):  <think> reasoning
process about different candidate answers here </think> <answer1> candidate_answer_1
</answer1> <confidence1> confidence level for candidate 1 here (number between 0 and
1) </confidence1> ...  exactly {K} pairs ...
```

.

---

**RLVR Multi Prompt**

```
A conversation between User and Assistant.  The user asks a question, and the
Assistant solves it.  You must propose multiple possible answers, not just one.
For each candidate, think separately about why it could be correct or incorrect.
Output EXACTLY {K} DISTINCT candidate answers.  FORMAT ONLY (no extra text):
<think> reasoning process about different candidate answers here </think> <answer1>
candidate_answer_1 </answer1> <answer2> candidate_answer_2 </answer2> ...  exactly
{K} answers ...
```

---

## C. Additional Training Details

We sample 32 responses per prompt using a temperature of 0.7 and train with an effective batch size of 1536. Optimization uses a constant learning rate with linear warmup, a base learning rate of $1 \times 10^{-6}$, and a warmup ratio of 0.05. All experiments are conducted on NVIDIA A100 and H100 GPUs, and we observe consistent performance trends across hardware types.

Following prior work (Turtel et al., 2025), we remove the standard deviation normalization from the advantage computation, which can improve learning stability in the presence of extreme miscalibration. Training is performed with the BNPO objective, which aggregates token-level losses normalized by the number of active tokens in each local training batch (Xiao et al., 2025). For both datasets, we set a maximum completion length of 1536. We do 1 epoch of training. System prompts for RLCR Single, RLCR Multi, RLVR Single, and RLVR Multi are in Appendix B.

**Format Reward:** We use a format reward to encourage adherence to the structured format required by the system prompts. In RLVR Single, models must format their output in `<think>` and `<answer>` tags. In RLCR Single, in addition to `<think>` and `<answer>` tags, we require a `<confidence>` tag for verbalized confidence. RLVR Multi requires a `<think>` tag and then $k$ `<answeri>` tags. RLCR Multi requires a `<think>` tag and then $k$ sets of `<answeri>` and `<confidencei>` tags. A valid response must contain all these tags in the correct order. Both format and calibration rewards are weighted equally.

In the Multi-RLCR and Multi-RLVR setting, we enforce the uniqueness required by the prompt by zeroing out all rewards

if the normalized answers extracted from the `<answeri>` tags are not unique. Additionally, in the N = 1 setting where our dataset assigns only one correct answer to each question, the format reward zeroes out if the sum of the confidences extracted from the `<confidencei>` tags is more than 1.

## D. Token Efficiency Comparison: Single Answer vs. Multi Answer RL

Here, we provide a visual of the average token compute usage of a set of answers generated by RLVR Single, RLVR Multi, RLCR Single, and RLCR Multi. As one can see, Multi-Answer training *significantly* increases compute efficiency.

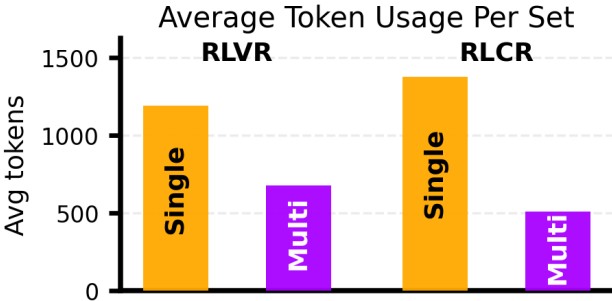

*Figure 7.* Average token usage for RLVR and RLCR in single and multi settings on DDXPlus.

## E. Unique Answer Analysis: RLVR Single vs. RLVR Multi

We create a Word Cloud comparing the answer diversity of RLVR Single and RLVR Multi on an example medical question. We run RLVR Single 30 times and RLVR Multi 10 times, with $k = 3$ answers per set. As one can see, RLVR single, even though it also collects 30 answers, only collects 3 unique answers. RLVR Multi, on the other hand, admits significantly more.

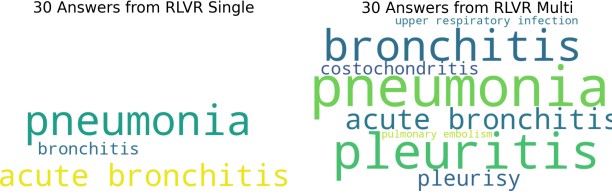

*Figure 8.* Word Cloud comparing Unique Answers gathered by RLVR Single and RLVR Multi

## F. Increasing $k$ in RLVR-Multi

| k | Evaluation Coverage, RLVR Multi |
|---|---|
| 2 | 0.78 |
| 3 | 0.68 |
| 4 | 0.65 |
| 5 | 0.62 |

*Table 3.* We report coverage, or the average number of correct answers generated in a set, on an evaluation set of $5,000$ questions as $k$ increases.

## G. Full Example - Medical RLVR Single, RLVR Multi, RLCR Single, RLCR Multi

**Question**

```
================================================================================
SYMPTOMS AND ANTECEDENTS:
================================================================================
Symptoms:
  1. Symptom: Have you been coughing up blood? -> Yes
  2. Symptom: Do you have pain somewhere, related to your reason for consulting? ->
    Yes
  3. Symptom: Characterize your pain:: a knife stroke
  4. Symptom: Do you feel pain somewhere?: lower chest
  5. Symptom: Do you feel pain somewhere?: posterior chest wall(R)
  6. Symptom: Do you feel pain somewhere?: posterior chest wall(L)
  7. Symptom: How intense is the pain?: 5
  8. Symptom: Does the pain radiate to another location?: nowhere
  9. Symptom: How precisely is the pain located?: 6
  10. Symptom: How fast did the pain appear?: 5
  11. Symptom: Are you experiencing shortness of breath or difficulty breathing in a
    significant way? -> Yes
  12. Symptom: Do you have a fever (either felt or measured with a thermometer)? ->
    Yes
  13. Symptom: Have you had chills or shivers? -> Yes
  14. Symptom: Do you have any lesions, redness or problems on your skin that you
    believe are related to the condition you are consulting for? -> Yes
  15. Symptom: What color is the rash?: pink
  16. Symptom: Do your lesions peel off?: Y
  17. Symptom: Is the rash swollen?: 1
  18. Symptom: Where is the affected region located?: commissure(L)
  19. Symptom: How intense is the pain caused by the rash?: 0
  20. Symptom: Is the lesion (or are the lesions) larger than 1cm?: N
  21. Symptom: How severe is the itching?: 0
  22. Symptom: Do you have diffuse (widespread) muscle pain? -> Yes
  23. Symptom: Have you noticed any new fatigue, generalized and vague discomfort,
    diffuse (widespread) muscle aches or a change in your general well-being related
    to your consultation today? -> Yes
  24. Symptom: Do you have a cough? -> Yes

Antecedents:
  1. Antecedent: Do you drink alcohol excessively or do you have an addiction to
    alcohol? -> Yes
  2. Antecedent: Do you have heart failure? -> Yes
  3. Antecedent: Do you have a chronic obstructive pulmonary disease (COPD)? -> Yes
  4. Antecedent: Have you traveled out of the country in the last 4 weeks?: N
  5. Antecedent: Is your BMI less than 18.5, or are you underweight? -> Yes

----------------------------------------------------------------
CORRECT / GOLD ANSWERS:
  1. Pneumonia
  2. Pulmonary neoplasm
  3. Bronchitis
  4. Tuberculosis
  5. Possible NSTEMI / STEMI
  6. GERD
  7. Unstable angina
  8. Pericarditis
  9. Stable angina
```

---

**RLVR Multi**

Okay, let's tackle this case. The patient is a 61-year-old male with several symptoms. He's coughing up blood, which is a red flag. He has chest pain described as a knife stroke, localized to the lower chest and right posterior chest wall. The pain is severe and sudden. He also has shortness of breath, fever, chills, and a rash on the left commissure that's pink and peeling. Additionally, he has diffuse muscle pain, fatigue, and a history of alcohol abuse, heart failure, COPD, and being underweight.

First, the hemoptysis (coughing up blood) and chest pain could point to a pulmonary issue. Given his COPD, maybe something like pulmonary embolism? But he also has heart failure, which could contribute to pulmonary issues. However, the sudden onset of severe pain and the presence of fever and chills might suggest an infection, like pneumonia. But the rash is another clue. The rash is on the commissure, pink, peeling, and swollen. That sounds like a fungal infection, maybe oral thrush, but the location is a bit unclear. Alternatively, could it be a drug reaction? He's on alcohol and has heart failure, so maybe medications are involved. But the main issues are the chest pain and hemoptysis.

Another possibility is tuberculosis, especially with the fever, night sweats (though not mentioned here), and hemoptysis. But the rash doesn't fit TB. Alternatively, a fungal infection like histoplasmosis, but that's more common in certain regions. The rash might be a secondary infection. Wait, the rash is on the commissure, which could be a sign of a fungal infection like candidiasis, but that's more oral. Maybe it's a skin manifestation of something else.

Considering the heart failure and COPD, could there be a pulmonary infection like pneumonia? The fever, chills, and shortness of breath support that. But the hemoptysis is more concerning for something like a pulmonary embolism or a tumor. However, the sudden onset and severe pain might point to a pulmonary embolism. But the rash is confusing. Alternatively, maybe the rash is a sign of a systemic infection, like sepsis, but that's a broader picture.

Another angle: the patient's alcohol use and underweight status could predispose him to infections or nutritional deficiencies. The diffuse muscle pain and fatigue could be due to systemic illness. The combination of hemoptysis, chest pain, fever, and rash makes me think of a fungal infection like coccidioidomycosis or histoplasmosis, but those usually have other symptoms. Alternatively, maybe a drug-induced reaction, but without knowing medications, it's hard. So the top three would be pulmonary embolism, pneumonia, and tuberculosis. Wait, the rash might be a clue for something else. Alternatively, the rash could be a sign of a systemic fungal infection. So maybe the three are pulmonary embolism, pneumonia, and fungal infection (like histoplasmosis). But the user wants three distinct diagnoses. Let me check again. The key symptoms are hemoptysis, chest pain, fever, chills, and rash. So possible diagnoses: 1. Pulmonary embolism, 2. Pneumonia, 3. Tuberculosis. Alternatively, 3. Fungal infection. But the rash might be more specific. Maybe the rash is a sign of a drug reaction, but without knowing medications, it's hard. So I'll go with pulmonary embolism, pneumonia, and tuberculosis.
\texttt{</think>}

\texttt{<answer1>} Pulmonary Embolism \texttt{</answer1>}
\texttt{<answer2>} Pneumonia \texttt{</answer2>}
\texttt{<answer3>} Tuberculosis \texttt{</answer3>}

## RLCR Multi

<think> Okay, let's tackle this case. The patient is a 61-year-old male with several symptoms. He's coughing up blood, which is a red flag. He has chest pain described as a knife stroke, localized to the lower chest and right posterior chest wall. The pain is severe (intensity 5) and sudden (appeared quickly). He also has shortness of breath, fever, chills, and a rash on the left commissure that's pink, peeling, and swollen. Additionally, he has diffuse muscle pain, fatigue, and a history of alcohol addiction, heart failure, COPD, and being underweight.

First, the presence of hemoptysis (coughing up blood) and chest pain raises concerns for pulmonary issues. Given his COPD, maybe a pulmonary embolism? But he also has heart failure, which could contribute to pulmonary congestion. However, the sudden onset of pain and fever might point towards an infection or embolism. The rash is on the commissure, which is the corner of the mouth, and it's pink, peeling, and swollen. That could be a fungal infection like oral thrush, but given his alcohol use and underweight status, maybe something else. Wait, the rash is on the commissure, and he has a fever, which could indicate an infectious process. The diffuse muscle pain and fatigue could be part of a systemic infection or a viral illness. However, the combination of hemoptysis, chest pain, fever, and chills suggests a more serious condition.

Pulmonary embolism (PE) is a possibility, especially with the sudden onset of chest pain and shortness of breath. But PE usually doesn't present with hemoptysis unless it's large. Alternatively, pneumonia could explain the fever, chills, and cough, but the hemoptysis is more concerning. Another angle: tuberculosis? But TB is more chronic. The rash might be a sign of a fungal infection, but I'm not sure. His history of heart failure could lead to pulmonary edema, which might cause hemoptysis, but the pain and fever are less typical.

Another thought: maybe a lung abscess or infection. The sudden onset and systemic symptoms (fever, chills) could point towards an infection. However, the rash is localized to the mouth, which might be a separate issue.

So, the top differentials would be pulmonary embolism, pneumonia, and maybe tuberculosis. But given the sudden onset, PE is higher. However, the presence of hemoptysis and chest pain could also suggest a pulmonary issue like a ruptured lung cyst or a malignancy. But with the fever and chills, infection is more likely. Alternatively, maybe a viral infection with secondary bacterial pneumonia.

Another possibility is acute bronchitis, but that's less likely with hemoptysis. The diffuse muscle pain and fatigue could be part of a systemic infection. So, the most likely diagnoses are pulmonary embolism, pneumonia, and tuberculosis. But I need to rank them. Given the sudden onset of chest pain, shortness of breath, fever, and hemoptysis, pulmonary embolism is a strong candidate. Pneumonia could also fit, especially with the fever and chills. Tuberculosis is less likely without a chronic cough or weight loss, but he is underweight. However, the rash might be a red herring.

So the three main possibilities are pulmonary embolism, pneumonia, and tuberculosis. But I need to check if there's a better fit. Alternatively, acute myocardial infarction? But the pain is described as a knife stroke, which could be pleuritic, pointing more to a pulmonary issue. So I'll go with pulmonary embolism, pneumonia, and tuberculosis as the top three. </think>

<answer1> Pulmonary Embolism </answer1> <confidence1> 0.45 </confidence1>

<answer2> Pneumonia </answer2> <confidence2> 0.35 </confidence2>

<answer3> Tuberculosis </answer3> <confidence3> 0.20 </confidence3>

---

**RLVR Single**

RLVR SINGLE:
\texttt{<think>}
Okay, let's tackle this medical case. The patient is a 61-year-old male with several
    symptoms. He's coughing up blood, which is a red flag. He has chest pain described
    as a knife stroke, localized to the lower chest and posterior walls on both sides.
    The pain is severe (intensity 5) and doesn't radiate. He also reports shortness of
    breath, fever, chills, and a pink rash that peels on the left commissure.
    Additionally, he has widespread muscle pain, fatigue, and a cough worse at night.
    His medical history includes alcohol addiction, heart failure, COPD, and being
    underweight.

First, coughing up blood (hemoptysis) along with chest pain and respiratory symptoms
    makes me think of pulmonary issues. The presence of fever, chills, and night
    sweats could indicate an infection. Given his COPD, he's already at risk for
    respiratory infections. But the combination of hemoptysis, fever, and chest pain
    might point towards something more serious like tuberculosis, especially since
    he's underweight and has a history of alcohol use, which can weaken the immune
    system. However, tuberculosis typically presents with chronic symptoms, though
    acute cases can occur.

Another possibility is pneumonia, but hemoptysis is less common in typical pneumonia.
    Then there's the rash-pink, peeling on the commissure. That could be a sign of a
    systemic infection or an autoimmune condition. However, the systemic symptoms like
    fever, muscle pain, and fatigue align more with an infectious process.

His heart failure could contribute to fluid retention and make him more susceptible
    to infections. The combination of hemoptysis, fever, and chest pain also makes me
    consider lung cancer, but that's more common in smokers and might present with
    different symptoms like weight loss and chronic cough. However, the acute onset of
    symptoms and the presence of fever might lean more towards an infection.

Tuberculosis should be considered, especially given the chronic nature of some
    symptoms, systemic involvement, and the patient's risk factors. The rash might be
    a secondary manifestation. Alternatively, fungal infections could be a
    possibility, especially in someone with COPD and immunocompromised state. But
    tuberculosis is more likely given the symptoms.

I should also check for other causes like bronchiectasis, which can cause hemoptysis
    and recurrent infections. But again, the fever and systemic symptoms point towards
    an active infection. Considering all factors, the most probable diagnosis is
    tuberculosis, given the combination of hemoptysis, fever, night sweats, weight
    loss, and the presence of a rash as a possible extrapulmonary manifestation.
    \texttt{</think>} \texttt{<answer>}Tuberculosis
\texttt{</answer>}
\texttt{<confidence>}95.\texttt{</confidence>}

---

**RLCR Single**

```
RLCR SINGLE (prefer correct; otherwise idx=0):
\texttt{<think>}
Okay, let's tackle this medical case. The patient is a 61-year-old male with several
    symptoms and antecedents. First, the key symptoms include coughing up blood, chest
    pain described as a knife stroke, shortness of breath, fever, chills, a pink rash
    that peels, muscle pain, fatigue, cough, night symptoms, and pain when inhaling.
    The antecedents are alcohol addiction, heart failure, COPD, and being underweight.

Coughing up blood (hemoptysis) is a red flag. Combined with chest pain and shortness
    of breath, this could point to a respiratory issue. The pain being a knife stroke
    suggests sharp, localized pain, which might indicate something like a pulmonary
    embolism (PE) or pleuritis. However, the presence of fever, chills, and a rash
    complicates things. The rash is pink, peels, and is on the commissure, which might
    be related to a systemic condition.

The patient has a history of COPD and heart failure, which are risk factors for PE.
    Alcohol use and underweight status could contribute to immune suppression or
    clotting issues. The diffuse muscle pain and fatigue could be due to a systemic
    infection or a condition like sepsis. However, the combination of hemoptysis,
    chest pain, and respiratory symptoms might lean towards tuberculosis or lung
    cancer, especially in someone with a history of smoking (though not mentioned
    here). Wait, the antecedents don't mention smoking, but COPD could be related.

I need to consider the most likely diagnosis. Given the symptoms, pulmonary embolism
    seems possible, especially with the history of COPD and heart failure. However,
    the rash and systemic symptoms might point towards something else like a systemic
    infection or even a drug reaction. Alternatively, tuberculosis could present with
    hemoptysis, fever, and night sweats, but the rash is less typical. Another angle:
    the patient's underweight status and alcohol use could lead to immunocompromise,
    making infections more likely. But the acute onset of symptoms (pain appearing
    quickly) might favor PE over a chronic condition like TB.

I should also check if the symptoms align with other conditions. The diffuse muscle
    pain and fatigue could be part of a systemic illness. The combination of
    hemoptysis, fever, chills, and chest pain is very concerning for PE. However,
    without imaging or more specific tests, it's challenging. The answer might be
    pulmonary embolism, but I'm not entirely sure due to the presence of the rash and
    other symptoms. Could it be something else like pneumonia with a rash? Or maybe a
    severe infection like sepsis with multi-system involvement. But the key symptom of
    hemoptysis points towards a pulmonary cause. I think PE is the most likely, but
    there's uncertainty due to the rash and other symptoms.
\texttt{</think>}

\texttt{<answer>} pulmonary embolism \texttt{</answer>}
\texttt{<analysis>} The diagnosis of pulmonary embolism (PE) is strongly suggested by
    hemoptysis, chest pain with a knife-like character, shortness of breath, fever,
    chills, and a history of COPD and heart failure, all of which increase PE risk.
    However, the presence of a pink, peeling rash on the commissure and diffuse muscle
    pain introduces ambiguity. These could indicate an alternative diagnosis such as a
    systemic infection, drug reaction, or tuberculosis. The patient's underweight
    status and alcohol use may contribute to immune compromise, complicating
    differential diagnosis. While PE remains the most likely explanation given the
    acute onset and respiratory symptoms, the rash and systemic manifestations
    introduce uncertainty. \texttt{</analysis>}
\texttt{<confidence>} 0.75 \texttt{</confidence>}
```

