# OpenReview forum: "Escaping the Mode: Multi-Answer Reinforcement Learning in LMs"
_ICML.cc/2026/Conference — ICML 2026 regular_

### Official Review · Reviewer_i35N · 2026-03-10

**Soundness:** 3
**Presentation:** 3
**Significance:** 3
**Originality:** 3
**Overall Recommendation:** 4
**Confidence:** 3

**Summary:**

This manuscript proposes Multi-Answer Reinforcement Learning, a framework that trains language models to generate sets of plausible answers with associated confidence scores rather than a single most likely response. Authors intends to address a central concept by introducing set-level reinforcement learning objectives and a calibrated reward variant that encourages both coverage of multiple correct answers and calibrated confidence estimates. Experiments on medical diagnosis and question answering datasets show improvements in recall, answer diversity, and token efficiency compared with single-answer RL baselines.

**Compliance With Llm Reviewing Policy:**

Affirmed.

**Key Questions For Authors:**

Please refer to weakness part.

**Limitations:**

yes

**Strengths And Weaknesses:**

Strengths
+ Authors identify an important limitation of standard RL training for LMs
+ The formulation is easy to understand and integrates naturally with existing RLVR/RLCR frameworks.
+ The extension to Multi-RLCR is valuable.
+ Experiments demonstrate consistent improvements in multiple metrics.

Weaknesses
- The evaluation is restricted to two QA-style datasets, both relatively narrow domains. It remains unclear whether the method generalizes to other tasks.
- Comparisons are limited. The proposed method mainly compares against: RLVR single-answer models or prompted multi-answer baselines. However, it does not compare with other diversity methods.
- While recall improves, top-1 accuracy does not consistently improve. This need fully analyze.
- The method requires models to output structured answer lists and confidence tags. I wonder whether the proposed method can be easily extended to other tasks.
- It unclear that the proposed method address mode collapse. It forces multiple outputs, which doesn't truely solve the collapse problem. Does the model's internal probability distribution become more diverse?

---

> ### Author Rebuttal · Authors · 2026-03-31
>
> Thank you for your thoughtful review. Below, we address your individual concerns.
>
> ---
>
> **New Dataset: Coding**
>
> To address these concerns, we trained several new models in the **coding** domain. We point the reviewer to the first section of our response to Reviewer G7Bn. Our results now show consistent improvements across three structurally diverse benchmarks - spanning inherently multi-answer, ambiguous single-answer, *and structured code generation* - that we believe significantly strengthen the generality of our conclusions.
>
> ---
>
> **Baselines**
>
> Thank you for raising this - we want to clarify that our evaluation doesinclude direct comparison with an inference-time diversity method. Specifically, we compare against *verbalized sampling*, a prompting-based approach that explicitly elicits diverse responses from a model at inference time. We welcome the reviewer's suggestions for additional baselines they believe would be most informative.
>
> ---
>
> **Top 1 Performance**
>
> This is a fair concern, and the results do show a nuanced picture that varies by task type.
>
> On new MBPP (coding dataset), the most deterministic of our benchmarks, Multi-Answer RL actually improves Top-1 accuracy by over 50% (RLVR Multi: 0.49 vs. RLVR Single: 0.29). We believe joint reasoning over multiple candidates encourages more thorough solution exploration before committing, benefiting the top answer alongside coverage. On HotPotQA-Modified, Top-1 is essentially identical across single and multi-answer methods.
>
> The one setting with a modest Top-1 trade-off is DDXPlus, where RLVR Single (0.50) outperforms RLVR Multi (0.42). However, DDXPlus has up to 9 simultaneously correct diagnoses - optimizing for a single mode is a fundamental mismatch with the task, and coverage is the more appropriate metric here. The trade-off is thus an expected consequence of training the model not to overcommit when genuine uncertainty exists.
>
> Overall, Multi-Answer RL matches or exceeds single-answer baselines on Top-1 in two of three settings and leads substantially on the most deterministic benchmark. We acknowledge the DDXPlus trade-off in our limitations and agree it is a worthwhile direction for future study, particularly in settings where top-1 accuracy on highly constrained tasks is the primary objective.
>
> ---
>
> **Other, non-verifiable tasks**
>
> We point the reviewer to the last section of our response to Reviewer G7Bn. We show that when evaluated on non-verifiable tasks, Multi-Answer RL consistently produces more diverse outputs than repeated single-answer sampling.
>
> ---
>
> **Addressing Diversity**
>
> We evaluate diversity using metrics that directly capture what matters: coverage (the number of *distinct* correct answers recovered per example), uniqueness, and embedding-based semantic distance on open-ended tasks. These metrics confirm that Multi-Answer RL produces outputs that are more diverse, more correct, and more semantically distinct than repeated single-answer sampling - this addresses mode collapse at the level that matters for downstream utility.
>
> Note that measures of diversity at the internal probability distribution level (such as entropy) are not necessarily aligned to diversity at the question level. For example, given a set of possible answers {"A","B","C","D"} for a given question, a model that assigns 0.3 to token "A" and then distributes the rest of the probability to all the other tokens in the vocabulary would have higher entropy than a model that assigns the probabilities "A":0.3, "B": 0.25, "C": 0.25, "D:" 0.2 (and assigns 0 to all other tokens). However, the lower-entropy model would have more diversity from the perspective of the downstream task.
>
>
> ---
>
> Thank you again for your review. We sincerely hope these clarifications and updates address your concerns and strengthen your confidence in the work.

---

### Official Review · Reviewer_ndV1 · 2026-03-12

**Soundness:** 3
**Presentation:** 4
**Significance:** 3
**Originality:** 3
**Overall Recommendation:** 5
**Confidence:** 4

**Summary:**

In this paper, the authors tackle the problem of lack of diversity after RL training. They motivate by explaining what vanilla RL objectives like GRPO is optimizing for and then explain that doing so will eventually lead the model to produce a single kind of answer (lack of diversity). Furthermore, they explain how this could be undesirable in medical applications like differential diagnosis.

To address this issue, the authors propose Multi-Answer Reinforcement Learning with Verifiable Rewards (Multi-RLVR) which uses a set-level reward (basically asking the model to produce a set of correct answers) and Multi-Answer Reinforcement Learning with Calibrated Rewards (Multi-RLCR) which uses a calibration reward to ensure the model predicts accurately about its uncertainties. Through their proposed approaches, the authors demonstrate improved diversity, recall and accuracy (at a set-level) in QA and medical-diagnostic tasks. Furthermore, they showcase that models trained with their approach are token-efficient in comparison to other parallel sampling approaches like repeated sampling.

**Compliance With Llm Reviewing Policy:**

Affirmed.

**Final Justification:**

I acknowledge the rebuttal by the authors and I recommend an Accept!

**Key Questions For Authors:**

No Major Questions, just performing experiments to investigate the generalizability of the proposed approach to other tasks like MATH, agentic tasks (even in a simple setting) would further strengthen the utility of the proposed approach.

**Limitations:**

Yes

**Strengths And Weaknesses:**

Strengths:

1) The paper is well-written and motivated clearly and neatly. The writing is good, easy to read and the narrative is easy to follow. Overall, a good presentation.
2) The proposed method is simple and can be easily reproduced.
3) The experiments are clearly set up, the baselines are explained well and the results support the claims made in the paper.

Weaknesses:

1) The problem of mode-collapse is a problem present even in domains like MATH and other verifiable domains, where there can be many ways of arriving at the correct solution. The papers focus on very niche tasks and if the authors could conduct some experiments to showcase the proposed approach’s generalizability to such tasks discussed above, it would further strengthen the proposed approach.

Overall, I think it is a good paper, however the kind of tasks where the proposed approaches are used appears to be niche, questioning its generalizability.

---

> ### Author Rebuttal · Authors · 2026-03-31
>
> Thank you for your thoughtful review. Below, we address your individual concerns.
>
> ---
>
> **New Domains**
>
> To address these point, we trained several new models in the **coding** domain. We point the reviewer to the first section of our response to Reviewer G7Bn. Our results now show consistent improvements across three structurally diverse benchmarks - spanning inherently multi-answer, ambiguous single-answer, *and structured code generation* - that we believe significantly strengthen the generality of our conclusions.
>
> MBPP is structurally distinct from both DDXPlus (ambiguous multi-answer medical diagnosis) and HotPotQA-Modified (ambiguous single-answer QA): - it is a well-specified, low-ambiguity benchmark where multiple correct answers arise from distinct algorithmic approaches rather than epistemic uncertainty.
> Multi-Answer RL achieves strong gains here as well: Multi-Answer RLVR reaches a Top-1 accuracy of 0.49 (vs. 0.29 for Single RLVR) while cutting token usage by more than half (235 vs. 511 tokens). The average number of AST-unique correct generations is significantly higher in both RLVR-Multi and RLCR-Multi than RLVR-Single and RLCR-Single, respectively.
>
>
>
> The consistent improvements across three structurally diverse benchmarks - spanning inherently multi-answer, ambiguous single-answer, and structured code generation - significantly strengthen the generality of our conclusions.
>
>
> |Method|Avg # Correct (↑)|Diversity (↑)|Efficiency (↓)|Top-1 Acc. (↑)|
> |---|---|---|---|---|
> |RLVR (Single Loss + Single Prompt)|0.98|2.09|511.73|0.29|
> |RLVR (Single Loss + Multi Prompt)|0.95|1.93|695.13|0.35|
> |Zero-Shot (Multi Prompt) [RLVR]|0.91|1.94|664.78|0.34|
> |**Multi-Answer RLVR (Multi Loss + Multi Prompt) (Ours)**|**1.35**|**2.98**|**235.49**|**0.49**|
> |---|---|---|---|---|
> |RLCR (Single Loss + Single Prompt)|0.89|2.13|518.07|0.27|
> |RLCR (Single Loss + Multi Prompt)|0.92|1.90|724.89|0.35|
> |Zero-Shot (Multi Prompt) [RLCR]|0.97|2.02|677.93|0.37|
> |**Multi-Answer RLCR (Multi Loss + Multi Prompt) (Ours)**|**1.38**|**2.94**|**250.94**|**0.48**|
>
>
> |Method|Set ECE (↓)|Top-1 ECE (↓)|Top-k ECE (↓)|Top-1 Brier (↓)|Top-k Brier (↓)|
> |---|---|---|---|---|---|
> |RLVR Single|-|0.56|-|0.55|-|
> |RLCR Single|-|0.53|-|0.53|-|
> |---|---|---|---|---|---|
> |RLVR Multi|0.44|0.54|0.51|0.51|0.48|
> |**RLCR Multi**|**0.26**|**0.37**|**0.33**|**0.38**|**0.34**|
>
> ---
>
> We also tested the **Math** domain.
> We trained two new models: RLVR Single and RLVR Multi on BigMathHard, a dataset of unambiguous single-answer math problems. Rather than degrading performance, RLVR Multi (K=3) outperforms RLVR Single prompted three times (Pass@1: 0.46 vs. 0.41; Pass@3: 0.83 vs. 0.62). This suggests that joint reasoning over multiple candidates encourages more thorough solution exploration before committing, benefiting even the top answer. Rather than hallucinating to fill slots, the model learns to generate structurally distinct, valid solution paths.
>
> |Method|Pass@1|Pass@3|
> |-|-|-|
> |Qwen2.5-7B-Base|0.27|0.53|
> |RLVR Single ×3|0.41|0.62|
> |RLVR Multi (K=3)|0.46|0.83|
>
>
> ---
>
>
> Additionally, we test the generalization of our trained models to **Non Verifiable domains**.  We transferred RLVR Multi and Single models (trained on HotPotQA) to 100 open-ended joke prompts (no verifiable ground truth), prompting Single 3× and Multi to generate 3 jokes in one pass, with Qwen3-8B as baseline.
>
> - *Semantic diversity* (↑): RLVR Multi **71.85** · Base 64.50 · RLVR Single 58.13
>
> - *Lexical diversity / ROUGE-L* (↓): RLVR Multi **0.168** · Base 0.231 · RLVR Single 0.261
>
> We replicated using models trained on DDXPlus (K=4), comparing 4 Single samples vs. one Multi generation of 4 jokes:
>
> - *Semantic diversity* (↑): RLVR Multi **85.16** · Base 64.02 · RLVR Single 60.23
>
> - *Lexical diversity / ROUGE-L* (↓): RLVR Multi **0.093** · Base 0.237 · RLVR Single 0.249
>
> Across both replications — different training domains, K values, and no reward signal — Multi-Answer RL consistently produces more diverse outputs than repeated single-answer sampling and the untuned base model, suggesting distributional reasoning generalizes beyond verifiable settings.
>
>
>
> ---
>
> Thank you again for your review. We sincerely hope these clarifications and updates address your concerns and strengthen your confidence in the work.

---

> > ### Author Rebuttal · Reviewer_ndV1 · 2026-04-01
> >
> > Can the authors elaborate on the experimental setup done for the experiments on coding and Math. For the tasks discussed in the main paper, having Multi-Answer RLCR makes sense and intuitive. I am not able to clearly visualize how the response looks like having multiple code solutions and the corresponding confidence would look like. So, I ask the authors to elaborate on the experimental setup in detail for both Multi-Answer RLVR and Multi-Answer RLCR and give some examples of sample rollouts during training.
> >
> > [Update]
> >
> > I am satisfied with the examples the authors provide. However, it is not clear to me how one will select the correct response to serve to the user (for example in the RLVR-Multi-Answer setup for Math). However, I think this paper is overall good and hope the authors incorporate all the feedbacks given to them during the rebuttal. I will increase the score.

---

> > > ### Author Response · Authors · 2026-04-02
> > >
> > > Thank you for the question. We clarify the setup below.
> > >
> > > **Multi-Answer RLVR:** The model generates multiple solutions within a single generation, each producing a final answer in a dedicated `<answerK>` tag. The correctness reward counts the number of *unique correct* answers. In math, this is capped at 1. In coding, multiple distinct correct implementations can each earn a point.
> > >
> > > **Multi-Answer RLCR:** In addition to the RLVR-Multi correctness reward, the model also emits a `<confidenceK>` for each answer. The calibration reward is the Multi-Brier score=the average Brier reward across all candidates.
> > > In math, confidences must sum to ≤1. In coding, they may sum to >1.
> > >
> > > **Example rollouts:**
> > >
> > > ---
> > > ### RLVR-Multi example
> > > **Problem:** On a circular track, two people A and B run in opposite directions. A completes a lap in 80 seconds. They meet every 30 seconds. How long does B take to complete a lap?
> > > ```xml
> > > <think>
> > > Path 1: Let's define the speed of A as \( v_A \) and B as \( v_B \). The circumference of the track is \( C \). The time for A to complete a lap is 1 minute and 20 seconds, which is 80 seconds. Therefore, \( v_A = \frac{C}{80} \). Since A and B meet every 30 seconds, the combined speed \( v_A + v_B = \frac{C}{30} \). We can solve for \( v_B \) and then the time for B to complete a lap.
> > >
> > > \[v_B = \frac{C}{30} - \frac{C}{80} = \frac{4C - 3C}{120} = \frac{C}{120}\]
> > > Thus, the time for B to complete a lap is: \[t_B = \frac{C}{v_B} = \frac{C}{\frac{C}{120}} = 120 \text{ seconds}\]
> > > Final: 120
> > >
> > > Path 2: Assume the track length is such that it can be measured in a way that simplifies the problem. Let's denote the length of the track as 240 units. This is a common multiple of 80 and 30, making the calculations simpler.
> > >
> > > - For A: \(\text{Time per lap} = 80 \text{ seconds}\), so speed in units/second = \(\frac{240}{80} = 3 \text{ units/second}\).
> > > - For B: They meet every 30 seconds, so combined speed = \(\frac{240}{30} = 8 \text{ units/second}\).
> > >
> > > Since the combined speed is 8 units/second and A's speed is 3 units/second, B's speed must be \(8 - 3 = 5 \text{ units/second}\). Therefore, the time for B to complete one lap is:
> > > \[t_B = \frac{240}{5} = 48 \text{ seconds}\]
> > > Final: 48
> > >
> > > Path 3: Consider the relative motion. In 30 seconds, A runs 80/2 = 40 units and B must run 240 - 40 = 200 units to meet A. Therefore, B's speed is:
> > > \[v_B = \frac{200}{30} = \frac{200}{30} = \frac{20}{3} \text{ units/second}\]
> > > The time for B to complete a lap is:
> > > \[t_B = \frac{240}{\frac{20}{3}} = \frac{240 \times 3}{20} = 36 \times 3 = 72 \text{ seconds}\]
> > > Final: 72
> > > </think>
> > > <answer1>120</answer1>
> > > <answer2>48</answer2>
> > > <answer3>72</answer3>
> > > ```
> > >
> > > Answer2 (48) is correct→**correctness reward=1**.
> > >
> > > ---
> > > ### RLCR-Multi example
> > > **Problem:** Check whether a list contains consecutive numbers.
> > >
> > > ```xml
> > > <think>
> > > Three approaches:
> > > 1. Sort and check pairwise differences are all 1.
> > > 2. Use a set to ensure uniqueness; check max - min == len - 1.
> > > 3. Iterate and check each element is one more than the previous.
> > > </think>
> > > <answer1>
> > > def is_consecutive_sort(nums):
> > >     nums_sorted=sorted(nums)
> > >     for i in range(len(nums_sorted)-1):
> > >         if nums_sorted[i+1]-nums_sorted[i]!=1:return False
> > >     return True
> > > </answer1><confidence1>0.7</confidence1>
> > > <answer2>
> > > def is_consecutive_set(nums):
> > >     if len(nums)!=len(set(nums)):return False
> > >     return max(nums)-min(nums)==len(nums)-1
> > > </answer2><confidence2>0.6</confidence2>
> > > <answer3>
> > > def is_consecutive_iterate(nums):
> > >     prev=nums[0]
> > >     for num in nums[1:]:
> > >         if num-prev!=1:return False
> > >         prev=num
> > >     return True
> > > </answer3><confidence3>0.8</confidence3>
> > > ```
> > >
> > > All correct→**correctness reward=3/3**.
> > >
> > > $$R_{\text{Brier}}=1-\frac{(1-0.7)^2+(1-0.6)^2+(1-0.8)^2}{3}\approx0.903$$
> > >
> > > ---
> > > ### RLCR-Multi example
> > >
> > > **Problem:** Write a function to find number of even elements in the given list.
> > > ```xml
> > > <think>Three approaches: 1. Use a lambda with filter to select even numbers and count them. 2. Iterate through the list and increment a counter for each even element. 3. Attempt a functional-style iteration applying a lambda to update a count.</think>
> > > <answer1>
> > > def count_even_elements(lst):
> > >     return len(list(filter(lambda x:x%2==0,lst)))
> > > </answer1><confidence1>0.7</confidence1>
> > > <answer2>
> > > def count_even_elements(lst):
> > >     c=0
> > >     for x in lst:
> > >         if x%2==0:c+=1
> > >     return c
> > > </answer2><confidence2>0.6</confidence2>
> > > <answer3>
> > > def count_even_elements(lst):
> > >     even_count=0
> > >     lst.forEach(lambda x:even_count+1 if x%2==0 else None)
> > >     return even_count
> > > </answer3><confidence3>0.3</confidence3>
> > > ```
> > > 2 correct→**correctness reward=2/3**.
> > >
> > > $$R_{\text{Brier}}=1-\frac{(1-0.7)^2+(1-0.6)^2+(0-0.3)^2}{3}=0.886$$
> > >
> > > --
> > >
> > > Please let us know if you have any further questions. If you find our response satisfactory, we would be grateful if you could consider raising your score. Thanks again for your time and efforts!

---

### Official Review · Reviewer_DmU2 · 2026-03-13

**Soundness:** 3
**Presentation:** 3
**Significance:** 3
**Originality:** 3
**Overall Recommendation:** 4
**Confidence:** 3

**Summary:**

## Summary
The paper states how to enable large language models (LLMs) to escape the "mode collapse" trap induced by standard reinforcement learning (RL) when facing tasks with incomplete information, high ambiguity, or complex reasoning requirements (like medical diagnosis). Instead of collapsing to a single modal output, the goal is to train the model to generate a set of plausible candidate answers, along with well-calibrated confidence scores, within a single forward reasoning pass. The author illustrates that elegant internalization of diversity seeking exploration—traditionally achieved through prohibitively expensive multiple samplings at inference time—directly into the generative objective during the model training phase. To achieve this, the authors propose a "Multi-Answer Reinforcement Learning" (Multi-Answer RL) framework, comprising two progressive methods: Multi-RLVR (introducing set-level rewards and format constraints to generate multiple distinct answers) and Multi-RLCR (incorporating a calibration penalty based on the Brier score, a strictly proper scoring rule). Experiments conducted on the DDXPlus (medical diagnosis) and HotPotQA-Modified (ambiguous QA) datasets demonstrate that this approach significantly outperforms single-answer RL baselines in terms of recall, answer diversity, and uncertainty calibration, while achieving orders-of-magnitude improvements in token efficiency.

**Compliance With Llm Reviewing Policy:**

Affirmed.

**Final Justification:**

The new experiment further makes the paper solid. Authors has made a lot of effort to address my issue. Only two questions remained:
1. Which model did you use the these two experiments? The MBPP seems to be lower if it's Qwen3-8B-Base(technical report https://arxiv.org/html/2505.09388v1, the evaluation metric in it is 69.80)
2. How is AST-unique used? Does it involve using Python’s ast library to abstract code snippets and then checking their equivalence?

Overall, this is good work with clear intuitive and presentation. I will keep my positive score. If author can fully address my final remained issue, I would like to increase my score

**Key Questions For Authors:**

## Questions
1. In domains like math and code, a single problem often has fundamentally distinct solutions (differing in both reasoning trajectories and final formats). Can the proposed method genuinely generate $K$ fundamentally different strategies (like distinct algorithmic structures)? How can the authors prove the model isn't merely relying on trivial "textual paraphrasing" or "variable renaming" just to satisfy the required answer count?
2. If this method is applied to more open-ended generative tasks, how would the authors redesign the "Format Penalty" to ensure that the multiple outputs possess genuine cognitive divergence and independence? I recommend providing a quantitative analysis of the semantic similarity between answers (like using embedding distances) in the rebuttal.
3. Does the model possess the capability to dynamically determine the optimal number of candidate answers? For instance, can it automatically output just 1 answer when its confidence is extremely high, and multiple answers when facing high ambiguity? Please provide an ablation study on the sensitivity of the Calibration Error to different fixed values of $K$.
4. In the Multi-RLCR formulation, how exactly are the correctness reward and the Brier calibration penalty balanced? Is there a risk that an overly large penalty term might cause the model to conservatively output "egalitarian confidences" (like regressing to the mean) to avoid heavy penalties?
5. The experiments are currently limited to a single 8B-scale model (Qwen3-8B). Could the authors provide evidence or preliminary results showing that Multi-Answer RL generalizes robustly across different model families (like LLaMA-3) and scales effectively to different parameter sizes(like 1.7B or 3B)

**Limitations:**

Yes

**Strengths And Weaknesses:**

## Strengths:
- Strong Theoretical Grounding: The paper goes beyond heuristic prompt engineering by seamlessly integrating "Proper Scoring Rules" (specifically the Brier Score) from statistics into the RL reward function. This provides a rigorous mathematical and game-theoretic guarantee that incentivizes the LLM to "honestly" report its internal probability distribution.
- Exceptional Token Efficiency: By generating multiple hypotheses and their probabilities in a single forward pass, the method completely eliminates the redundant Chain-of-Thought (CoT) generation overhead associated with multiple independent rollouts. This holds immense practical value for latency-sensitive and compute-constrained real-world applications (like real-time clinical decision support).
- Clear Motivation & Practical Relevance: The "entropy collapse" caused by standard RLHF/RLVR blindly optimizing for the maximum expected reward is currently a core pain point in the alignment community. Defining and solving this problem in a high-stakes scenario like medical diagnosis—which inherently possesses "multi-solution" characteristics—is a highly precise and relevant entry point.
## Weaknesses:
- Lack of Evaluation on Math/Code Domains: The experiments are confined to knowledge-retrieval and classification datasets (QA and medical). However, the core testbeds for modern LLM reasoning are math (like MATH) and code generation (like HumanEval), which naturally feature "multiple valid trajectories" (like different algorithmic complexities or diverse proof paths). The absence of evaluation in these highly structured, logical reasoning domains weakens the generalizability of the proposed method.
- Static Dependency on Hyperparameter $K$: The system appears to require a statically defined $K$ (number of answers) in the prompt. If a question is objectively unambiguous and has only one valid answer, forcing the model to output multiple distinct answers may inevitably induce hallucinations and artificially flatten what should be a sharp, highly confident probability distribution.
- Scalability of Verification for Open-Ended Solutions: In diagnostic datasets, verifying an answer simply requires checking it against a predefined ground truth set $\mathcal{Y}^*$. However, in open-ended code or math tasks, the cost of verifying whether $K$ different solutions are all correct (like running multiple test cases or symbolic solvers) will scale exponentially. The paper does not discuss the scalability of this framework under high-cost verification scenarios.
- Potential Alignment Tax (Top-1 Performance Trade-off): Encouraging multi-answer exploration may distract the model's focus, potentially leading to a degradation in Top-1 accuracy on extremely simple, deterministic standard benchmarks compared to mode-seeking, single-answer baselines.

---

> ### Author Rebuttal · Authors · 2026-03-31
>
> Thank you for your thoughtful review. Below, we address your individual concerns.
>
> ---
> **New Coding Domain**
>
> We trained several new models in the **coding** domain. We point the reviewer to the first section of our response to Reviewer G7Bn. Our results now show consistent improvements across three structurally diverse benchmarks - inherently multi-answer, ambiguous single-answer, and structured code generation - that we believe significantly strengthen the generality of our conclusions.
>
> ---
> **Unambiguous / Single Answer domains**
>
> We directly tested a domain that has unambiguous, single answer questions: math. We trained RLVR Single and RLVR Multi k=3 on BigMathHard. Rather than degrading performance, RLVR Multi (K=3) outperforms RLVR Single prompted three times (Pass@1: 0.46 vs. 0.41; Pass@3: 0.83 vs. 0.62). This suggests that joint reasoning over multiple candidates encourages more thorough solution exploration before committing. Rather than uselessly hallucinating to fill slots, the model learns to generate structurally distinct possible solution paths.
> |Method|Pass@1|Pass@3|
> |-|-|-|
> |Qwen2.5-7B-Base|0.27|0.53|
> |RLVR Single ×3|0.41|0.62|
> |RLVR Multi (K=3)|0.46|0.83|
>
> ---
> **Verification Scalability for Open-Ended Solutions**
>
> Verification scales linearly in K, not exponentially. For math, correctness is checked via symbolic matching (math_verify), which runs in milliseconds/candidate. For code (MBPP), each candidate is executed in an isolated subprocess with a 3-sec timeout (worst-case K x 3 seconds). In practice, AST-level deduplication loweres cost by executing only unique programs. We note that checking cost also applies for sampled Best-of-K. Settings that require LLM-based judges per candidate may introduce scalability challenges, which we will discuss in the paper.
>
> ---
> **Top-1 Performance**
>
> We refer the reviewer to the Top 1 Performance section in our response to reviewer i35N.
>
> ---
> **Fundamental Diversity in Coding**
>
> In our coding experiments, we enforce AST-level uniqueness: candidates with identical abstract syntax trees are penalized, preventing variable renaming or superficial variation from satisfying the multi-answer requirement.
>
> The higher coverage of RLVR-Multi shows that models learn structurally distinct implementations rather than trivial variants. For example, on the simple task “check whether two numbers have the same number of digits,” RLVR-Single produces AST-identical solutions based on string length, while RLVR-Multi generates AST-unique approaches, including string-based comparison, iterative digit counting via integer division, and variants with different control flow.
>
> *RLVR Single Cands. 1,2,3*:
> ```python
> def same_number_of_digits(num1,num2):
>  return len(str(num1))==len(str(num2))
> ```
> *RLVR Multi*:
> ```python
> def same_number_of_digits(num1,num2):
>  return len(str(num1))==len(str(num2))
>
> def same_number_of_digits(num1,num2):
>  d1=d2=0
>  while num1>0:num1//=10;d1+=1
>  while num2>0:num2//=10;d2+=1
>  return d1==d2
>
> def same_number_of_digits(num1,num2):
>  s1,s2=str(num1),str(num2)
>  return len(s1)==len(s2)
> ```
> ---
> **Open Ended Tasks**
>
> We point the reviewer to the last section of our response to Reviewer G7Bn. We show that when evaluated on non-verifiable tasks, Multi-Answer models consistently produce more diverse outputs than repeated single-answer sampling.
>
> ---
> **Dynamic K**
>
> We fix K during training to prevent reward hacking under set-level objectives like RLCR-Multi. With variable K, the model can game rewards by padding with low-confidence candidates (inflating coverage without meaningful uncertainty) or diluting probability mass (with incorrect answers assigned very low probabilities) to artificially inflate calibration reward.
>
> For these reasons, we fix K during training, which enforces a consistent structure and requires the model to meaningfully allocate probability mass across candidates.
>
> To address your question directly, however, we trained RLCR-Multi on the medical domain with K = 2, 4, and 5. Brier scores were stable across settings (K=2: 0.21, K=4: 0.22, K=5: 0.21), and each model still produced diverse, well-differentiated confidences.
>
> ---
> **Multi-RLCR Reward**
>
> Correctness and Brier calibration rewards are weighted equally. On DDXPlus and MBPP, RLCR-Multi achieves substantially better set-level calibration than RLVR-Multi without collapsing to uniform confidence. The anomaly is HotPotQA-Modified, where the model develops a prior to sum confidences to 1 - rational but suboptimal in a hard single-label setting - which we discuss as a limitation.
>
> ---
> **Different Model Family**
>
> We trained RLVR Single and RLVR Multi on OLMo2-7B. RLVR Multi achieves coverage of 1.547 vs. 1.020 for RLVR Single (~52% gain), replicating the Qwen3-8B pattern and confirming that Multi-Answer RL benefits are not architecture-specific.
>
> ---
> Thank you again for your review. We sincerely hope these clarifications and updates address your concerns and strengthen your confidence in the work.

---

> > ### Author Rebuttal · Reviewer_DmU2 · 2026-04-04
> >
> > Thank you for authors' new experiment results. Can author systematically make all these new result to revised version to make the paper more persuasive? And also, can author test method in more challenging dataset like(Math,  codeforces). If the time is limited, just use a few samples to train is ok. Basically, I will keep my positive option on this paper, if there are more experiment result, I would like to increase my score.

---

> > > ### Author Response · Authors · 2026-04-05
> > >
> > > Thank you again for the helpful suggestion. While openreview doesn't allow us to update the paper during the rebuttal period, we will absolutely incorporate all of the new experimental results from the rebuttal into the final revised version of the paper to strengthen its empirical support.
> > >
> > > In particular, we **extended evaluation to more challenging settings in both math (BigMathHard) and coding (MBPP)**.
> > > - **On BigMathHard, RLVR-Multi (K=3) improves Pass@3 from 0.62 to 0.83 over RLVR-Single (and improves Pass@1 from 0.41 to 0.46).**
> > >
> > > - **On MBPP, Multi-Answer RL substantially improves Top-1 accuracy (0.49 vs. 0.29) while increasing the number of AST-unique correct solutions and reducing token usage by over 50%**.
> > >
> > > These results show that the approach not only maintains diversity but also strengthens correctness and efficiency in structured reasoning domains. We hope these additions address your concerns, and we would greatly appreciate your consideration of an updated score in light of this.

---

### Official Review · Reviewer_G7Bn · 2026-03-16

**Soundness:** 3
**Presentation:** 3
**Significance:** 3
**Originality:** 3
**Overall Recommendation:** 4
**Confidence:** 3

**Summary:**

This paper studies multi-answer reinforcement learning for language models. Instead of training the model to output only one best answer, the paper proposes training objectives that let the model generate a small set of plausible answers in one pass, and further extends this setup to produce calibrated confidence scores for each answer. Experiments on medical diagnosis and modified question answering benchmarks show improvements in answer coverage, diversity, calibration, and token efficiency compared with single-answer baselines and prompting-based alternatives.

**Compliance With Llm Reviewing Policy:**

Affirmed.

**Final Justification:**

This paper addresses a meaningful problem, especially for settings where multiple plausible answers and calibrated uncertainty are more appropriate than a single modal output. I found the formulation clean, the method easy to follow, and the empirical results broadly supportive of the main claims, so I view the work as technically sound and clearly presented. In terms of originality, I see it as a meaningful extension of the RLVR/RLCR line rather than a radically new direction, but still one that others are likely to build on. My initial concerns were mainly about the scope of the claims and the need to position the contribution more explicitly relative to prior single-answer RLVR/RLCR-style work. The rebuttal addressed these points well by adding results on an additional domain, clarifying sensitivity to K, and better articulating the conceptual move from single-answer to set-valued training. These responses strengthened my confidence in the paper, although they did not materially change my overall recommendation because my original assessment was already positive.

**Key Questions For Authors:**

1. How sensitive are the main results to the choice of K, especially in settings where the number of plausible answers is highly variable across examples?

2. Do the authors expect the same gains to hold on broader open-ended generation tasks where correct answer sets are less well defined and exact verification is harder?

**Limitations:**

yes

**Strengths And Weaknesses:**

The paper addresses a meaningful problem. The motivation is clear, especially for settings where multiple answers may be reasonable or where uncertainty should be represented more explicitly. The method is simple to understand, the formulation is clean, and the empirical results are generally consistent with the paper’s main claims. I also found the paper well-organized and easy to follow.

My main weaknesses are relatively minor. First, it would be helpful if the authors could clarify the intended scope of the claims, since the current evaluation is conducted across two specific settings and it is not yet clear how broadly the conclusions are expected to transfer. Second, it would strengthen the paper if the authors could more explicitly position the method relative to prior RLVR/RLCR-style work, especially to clarify what they see as the main conceptual step beyond a well-executed extension of existing ideas.

---

> ### Author Rebuttal · Authors · 2026-03-31
>
> Thank you for your thoughtful review. Below, we address your individual concerns.
>
> ---
> **New Dataset: Coding**
>
> We have extended our evaluation to a third domain: **coding (MBPP)**. We trained several new models that substantially broadens the scope of our claims. MBPP is structurally distinct from both DDXPlus (ambiguous multi-answer medical diagnosis) and HotPotQA-Modified (ambiguous single-answer QA): - it is a well-specified, low-ambiguity benchmark where multiple correct answers arise from distinct algorithmic approaches rather than epistemic uncertainty.
>
> Multi-Answer RL achieves strong gains here as well: Multi-Answer RLVR reaches a Top-1 accuracy of 0.49 (vs. 0.29 for Single RLVR) while cutting token usage by more than half (235 vs. 511 tokens). The average number of AST-unique correct generations is significantly higher in both RLVR-Multi and RLCR-Multi than RLVR-Single and RLCR-Single, respectively.
>
> The consistent improvements across three structurally diverse benchmarks - spanning inherently multi-answer, ambiguous single-answer, and structured code generation - significantly strengthen the generality of our conclusions.
> |Method|Avg # Correct (↑)|Diversity (↑)|Efficiency (↓)|Top-1 Acc. (↑)|
> |---|---|---|---|---|
> |RLVR (Single Loss + Single Prompt)|0.98|2.09|511.73|0.29|
> |RLVR (Single Loss + Multi Prompt)|0.95|1.93|695.13|0.35|
> |Zero-Shot (Multi Prompt) [RLVR]|0.91|1.94|664.78|0.34|
> |**Multi-Answer RLVR (Multi Loss + Multi Prompt) (Ours)**|**1.35**|**2.98**|**235.49**|**0.49**|
> |---|---|---|---|---|
> |RLCR (Single Loss + Single Prompt)|0.89|2.13|518.07|0.27|
> |RLCR (Single Loss + Multi Prompt)|0.92|1.90|724.89|0.35|
> |Zero-Shot (Multi Prompt) [RLCR]|0.97|2.02|677.93|0.37|
> |**Multi-Answer RLCR (Multi Loss + Multi Prompt) (Ours)**|**1.38**|**2.94**|**250.94**|**0.48**|
> &nbsp;
> |Method|Set ECE (↓)|Top-1 ECE (↓)|Top-k ECE (↓)|Top-1 Brier (↓)|Top-k Brier (↓)|
> |---|---|---|---|---|---|
> |RLVR Single|-|0.56|-|0.55|-|
> |RLCR Single|-|0.53|-|0.53|-|
> |---|---|---|---|---|---|
> |RLVR Multi|0.44|0.54|0.51|0.51|0.48|
> |**RLCR Multi**|**0.26**|**0.37**|**0.33**|**0.38**|**0.34**|
>
> ---
>
> **Positioning Relative to Prior RLVR/RLCR Work**
>
> The core contribution beyond RLVR and RLCR is internalizing distributional reasoning into a single generation. Prior RLVR collapses onto a single mode; RLCR adds calibration but stays in the single-answer regime. Multi-Answer RL is the first approach to train models to explicitly recover the answer distribution in one generation - qualitatively different from repeated sampling, which re-instantiates the same dominant trace (Figure 5). It strictly subsumes both single-answer methods as special cases (Section 3.1, cases 1–4). The key shift is from optimizing a single high-reward completion to optimizing a set-valued output that explicitly covers model uncertainty.
>
> ---
>
> **Effects of K.**
>
> We trained separate RLVR-Multi models for K = 2–5 on DDXPlus, where gold diagnosis counts vary widely per example. Average coverage on a held-out set of 5,000 questions:
> - K=2: 0.65 · K=3: 0.79 · K=4: 0.87 · K=5: 1.25 Coverage increases monotonically and training remains stable across all K (Figure 6). Since outputs are constrained to be unique, gains reflect the model surfacing additional correct hypotheses, not repeating dominant answers. The results suggest our approach is robust to variable ground-truth set sizes.
>
> ---
>
> **Open-ended tasks**
>
> We transferred RLVR Multi and Single models (trained on HotPotQA) to 100 open-ended joke prompts (no verifiable ground truth), prompting Single 3× and Multi to generate 3 jokes in one pass, with Qwen3-8B as baseline.
>
> - *Semantic diversity* (↑): RLVR Multi **71.85** · Base 64.50 · RLVR Single 58.13
>
> - *Lexical diversity / ROUGE-L* (↓): RLVR Multi **0.168** · Base 0.231 · RLVR Single 0.261
>
> We replicated using models trained on DDXPlus (K=4), comparing 4 Single samples vs. one Multi generation of 4 jokes:
> - *Semantic diversity* (↑): RLVR Multi **85.16** · Base 64.02 · RLVR Single 60.23
>
> - *Lexical diversity / ROUGE-L* (↓): RLVR Multi **0.093** · Base 0.237 · RLVR Single 0.249
>
> Across both replications - different training domains, K values, and no reward signal - Multi-Answer RL consistently produces more diverse outputs than repeated single-answer sampling and the untuned base model, suggesting distributional reasoning generalizes beyond verifiable settings.
>
> ---
>
> Thank you again for your review. We sincerely hope these clarifications and updates address your concerns and strengthen your confidence in the work.

---

> > ### Author Rebuttal · Reviewer_G7Bn · 2026-04-02
> >
> > Thank you for the thorough rebuttal and the additional experiments on coding, math, K sensitivity, and broader generalization. These clarifications address my main questions and strengthen the paper. I am keeping my score unchanged because my original score already reflected a positive assessment under the current scope.

---

> > > ### Author Response · Authors · 2026-04-02
> > >
> > > Thank you again for the thoughtful follow-up. Since our rebuttal and new experiments seem to have addressed your main concerns, we would be very grateful if you would consider updating your score to better reflect your current assessment.

---

### Decision · Program_Chairs · 2026-04-30

**Decision:**

Accept (regular)

**Comment:**

The paper proposes a multi-answer reinforcement learning framework for large language models that trains models to generate a set of plausible answers in a single forward pass. The reviewers agree that the work addresses an important limitation of standard RLVR-style training, is clearly written, and is technically sound, with empirical results that substantiate the main claims. After the rebuttal, all the reviewers recommended acceptance.